EMBO
Molecular Medicine

# Parkin is a disease modifier in the mutant SOD1 mouse model of ALS

Gloria M Palomo[1], Veronica Granatiero[1], Hibiki Kawamata[1], Csaba Konrad[1], Michelle Kim[1], Andrea J Arreguin[1], Dazhi Zhao[1], Teresa A Milner[1,2] & Giovanni Manfredi[1,*]

## Abstract

Mutant Cu/Zn superoxide dismutase (SOD1) causes mitochondrial alterations that contribute to motor neuron demise in amyotrophic lateral sclerosis (ALS). When mitochondria are damaged, cells activate mitochondria quality control (MQC) mechanisms leading to mitophagy. Here, we show that in the spinal cord of G93A mutant SOD1 transgenic mice (SOD1-G93A mice), the autophagy receptor p62 is recruited to mitochondria and mitophagy is activated. Furthermore, the mitochondrial ubiquitin ligase Parkin and mitochondrial dynamics proteins, such as Miro1, and Mfn2, which are ubiquitinated by Parkin, and the mitochondrial biogenesis regulator PGC1α are depleted. Unexpectedly, Parkin genetic ablation delays disease progression and prolongs survival in SOD1-G93A mice, as it slows down motor neuron loss and muscle denervation and attenuates the depletion of mitochondrial dynamics proteins and PGC1α. Our results indicate that Parkin is a disease modifier in ALS, because chronic Parkin-mediated MQC activation depletes mitochondrial dynamics-related proteins, inhibits mitochondrial biogenesis, and worsens mitochondrial dysfunction.

**Keywords** amyotrophic lateral sclerosis; mitochondria quality control; mitophagy; Parkin; SOD1

**Subject Categories** Genetics, Gene Therapy & Genetic Disease; Neuroscience

## Introduction

Amyotrophic lateral sclerosis (ALS) is a fatal neurodegenerative disease caused by the demise of upper and lower motor neurons. Symptoms include muscle weakness and atrophy, leading to fatal paralysis, usually within 5 years after onset (Kiernan *et al*, 2011). Approximately 90% of ALS cases have no family history, while the remaining 10% are due to inherited mutations. Of the numerous forms of inherited ALS (Bettencourt & Houlden, 2015), approximately 20% are caused by mutations in the gene Cu/Zn superoxide

dismutase (*SOD1*), which was the first ALS gene discovered (Rosen *et al*, 1993). Over 180 different mutations in *SOD1* have been identified and associated with the disease (Wroe *et al*, 2008). After the discovery of SOD1, many other genes have been associated with ALS (Ghasemi & Brown, 2017). These genes encode for proteins involved in different cellular pathways, from RNA metabolism, to nuclear transport, to protein and organellar quality control, highlighting the complexity of ALS pathogenesis (Peters *et al*, 2015). Numerous evidence of mitochondrial functional and morphological alterations indicates that these organelles are predominantly involved in ALS pathogenesis (Smith *et al*, 2017).

SOD1 is mostly localized in the cytosol, but a fraction of the protein localizes to the mitochondrial intermembrane space (IMS; Okado-Matsumoto & Fridovich, 2001). In mitochondria, misfolded and aggregated mutant SOD1 localizes in the IMS (Vijayvergiya *et al*, 2005; Kawamata & Manfredi, 2010) and on the outer membrane (OMM; Vande Velde *et al*, 2008), where it can establish aberrant interactions with proteins, such as Bcl2 (B-cell lymphoma 2; Pasinelli *et al*, 2004), VDAC1 (voltage-dependent anion-selective channel 1; Israelson *et al*, 2010), and the mitochondrial protein import machinery (Li *et al*, 2010). Mutant SOD1 causes defects of mitochondrial calcium homeostasis, energy production, axonal transport, and alter interactions with other organelles, such as the endoplasmic reticulum (Manfredi & Kawamata, 2016; Smith *et al*, 2017). Moreover, mitochondrial swelling, cristae disruption (Gould *et al*, 2006; Sasaki & Iwata, 2007), and enhanced presence of mitochondria in autophagic vacuoles (Sasaki, 2011) are striking pathological features that begin early on and chronically worsen throughout the disease course.

Maintenance of a sufficient pool of functional mitochondria is vital to neurons and depends on the coordinated activity of both mitochondrial biogenesis and mitochondrial quality control (MQC). MQC mechanisms include several processes, including selective mitochondrial autophagy (mitophagy; McWilliams & Muqit, 2017), targeted mitochondrial proteostasis (Quiros *et al*, 2015), and the generation of mitochondria-derived vesicles (Soubannier *et al*, 2012). The mechanism of selective tagging of damaged mitochondria by MQC has been largely ascribed to the coordinated action of PINK1 and Parkin (Mouton-Liger *et al*, 2017), resulting in phosphorylated ubiquitin in the OMM (Kane *et al*, 2014; Kazlauskaite *et al*,

1 Feil Family Brain and Mind Research Institute, Weill Cornell Medicine, New York, NY, USA
2 Harold and Margaret Milliken Hatch Laboratory of Neuroendocrinology, The Rockefeller University, New York, NY, USA
*Corresponding author. Tel: +1 646 962 8172; E-mail: gim2004@med.cornell.edu

2014; Koyano et al, 2014), which in turn recruits autophagy receptors to the mitochondrial surface (Okatsu et al, 2010; Wong & Holzbaur, 2014; Lazarou et al, 2015). Intriguingly, genes associated with ALS encode for proteins involved in autophagy, mitophagy, and proteostasis, including TBK1 (tank-binding kinase 1), and VCP (valosin-containing protein or p97; Johnson et al, 2010; Freischmidt et al, 2015), suggesting that MQC participates in disease pathogenesis. However, the role of MQC in the central nervous system subjected to chronic mitochondrial damage, such as in the case of mutant SOD1, has not been fully elucidated. While an acute insult to mitochondria activates MQC, which clears damaged mitochondria, chronic damage could result in prolonged activation of MQC and depletion of essential mitochondrial components due to excessive turnover.

Here, to understand the role of MQC in ALS, we investigate transgenic mice that express human SOD1 carrying the pathogenic G93A mutation (SOD1-G93A). We find increased MQC activity in the spinal cord of SOD1-G93A mice, resulting in enhanced Parkin turnover and depletion of mitochondrial components. We show that knocking out Parkin in SOD1-G93A mice leads to attenuation of MQC and slows down the depletion of mitochondrial dynamics-related proteins and mitochondrial function, resulting in delayed motor neuron loss, muscle denervation, and disease progression. Taken together, these findings demonstrate that Parkin-mediated MQC modifies the disease course in a preclinical animal model of ALS.

# Results

## MQC markers increase in spinal cord mitochondria of SOD1-G93A mice

Progressive defects of mitochondrial function and dynamics have been well documented in spinal cord motor neurons of SOD1-G93A mouse (Mattiazzi et al, 2002; Gould et al, 2006; Sasaki & Iwata, 2007). Therefore, we investigated whether mitochondrial damage induces MCQ activity in the spinal cord of SOD1-G93A mice in the B6SJL genetic background, which has an average lifespan of 129 days (Gurney et al, 1994). In this mouse, we analyzed the autophagosome adaptor p62 at 30 and 60 days of age (presymptomatic stage), 90 days (disease onset), and 120 days (symptomatic stage). Remarkably, mutations in p62 have been associated with familial ALS (Maruyama et al, 2010; Fecto et al, 2011). Levels of p62 (Fig 1A and B), normalized by the mitochondrial ATPase β-subunit (Complex V), were significantly increased in enriched mitochondrial fractions from spinal cord of SOD1-G93A mice at 120 days, suggesting that MQC activity was increased at the advanced disease stage. Notably, an increase in p62 associated with mitochondria was only observed in the SOD1-G93A spinal cord, and not in mitochondria from the spinal cord of transgenic mice expressing wild-type human SOD1 at the same age, indicating that the activation of MQC was not simply caused by SOD1 overexpression, but was specific to mutant SOD1.

Increased MQC can induce a decrease in mitochondrial content due to increased organellar turnover. Accordingly, the levels of two mitochondrial proteins, the protein translocator of the inner membrane, Tim23 (Fig 1C and D) and subunit 1 of cytochrome

oxidase, COX1 (Fig 1E and F), normalized by β-actin, were decreased in spinal cord homogenates of SOD1-G93A mice. The decrease in mitochondrial proteins was observed at 120 days, the same age at which p62 was increased in the SOD1-G93A spinal cord, and did not affect wild-type SOD1 mice. Since Tim23 and COX1 are inner membrane proteins that cannot be accessed by the proteasome for degradation, these results suggest that mitochondrial protein depletion in SOD1-G93A spinal cord is caused by increased organellar turnover.

The maintenance of the mitochondrial protein pool depends on turnover, but also on biogenesis. PGC1α is a transcription coactivator that regulates mitochondria biogenesis (Puigserver & Spiegelman, 2003) and controls its own transcription. Therefore, to assess the effects of SOD1-G93A on mitochondrial biogenesis, we measured PGC1α mRNA levels. PGC1α mRNA was decreased in SOD1-G93A spinal cord starting at 90 days of age (Fig 1G). This result is consistent with the previously reported decline of PGC1α in SOD1 mutant mice (Thau et al, 2012). Taken together, this evidence suggests that increased MCQ and decreased mitochondrial biogenesis contribute to a decline of mitochondrial pools in SOD1-G93A spinal cord.

## Mitophagy flux is increased in SOD1-G93A motor neurons

Mitochondria from spinal cord of SOD1-G93A mice show increased levels of p62 at 120 days, suggesting enhanced MQC activity, which results in mitochondrial depletion, possibly through mitophagy. Therefore, to test the hypothesis that mitophagy activity is increased in motor neurons, we crossed C57Bl6 SOD1-G93A mice with mice expressing the mitophagy reporter mt-Keima (Sun et al, 2015) to obtain double transgenic mt-Keima/SOD1-G93A mice. In the mt-Keima mouse, the ratiometric fluorescent protein Keima is targeted to the mitochondrial matrix. Keima fluorescence properties depend on pH. So, when mitochondria expressing mt-Keima are engulfed in autophagic vesicles and the environment becomes acidic, the excitation spectrum of the probe shifts, allowing for visualization and quantification of mitophagy (Katayama et al, 2011).

Mitophagy was assessed in mt-Keima/SOD1-G93A spinal cords at 60, 90, and 120 days of age. Motor neurons labeled with mt-Keima were identified by their morphology and their localization in the ventral spinal cord, anterior of the central canal (Fig 2A and B). This unique in vivo approach was highly informative, as it allowed us to visualize a time-dependent increase in acidic mitochondria, in both mt-Keima/Non Tg and mt-Keima/SOD1-G93A spinal cords (Fig 2C), indicating that mitophagy fluxes increase postnatally, as motor neurons mature. Furthermore, when we compared genotypes at each time point, we detected a significant increase in mitophagy at 60 and 90 days in mt-Keima/SOD1-G93A compared to mt-Keima/Non Tg. However, there was no significant difference between genotypes at 120 days. These results indicate that mitophagy is enhanced early on in SOD1-G93A motor neurons relative to controls, but no longer at the later disease stage, possibly because the MQC machinery becomes dysfunctional or depleted.

## Parkin is decreased in SOD1-G93A spinal cord

Mitochondrial translocation of Parkin is one of the early steps in MQC, leading to the recruitment of autophagy receptors to the

**A**

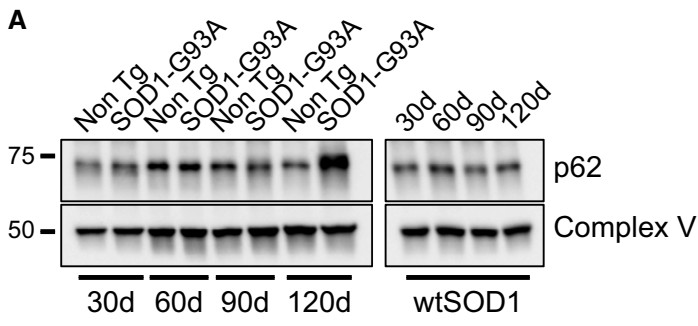

**B**

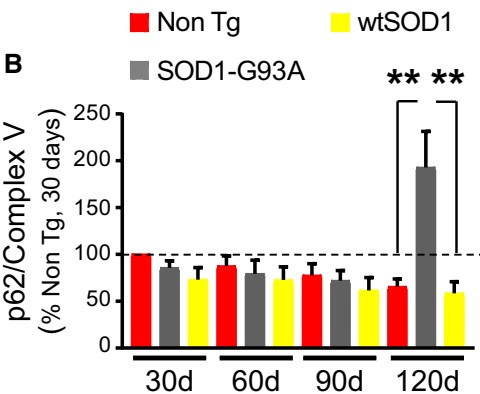

**C**

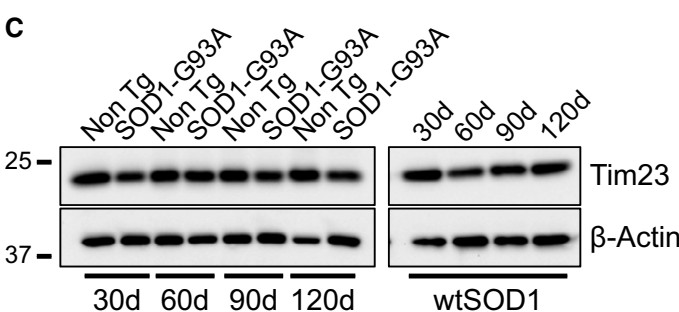

**D**

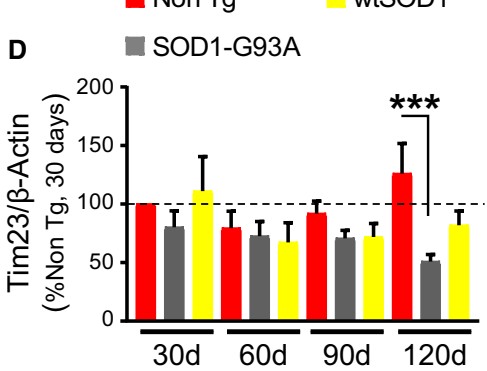

**E**

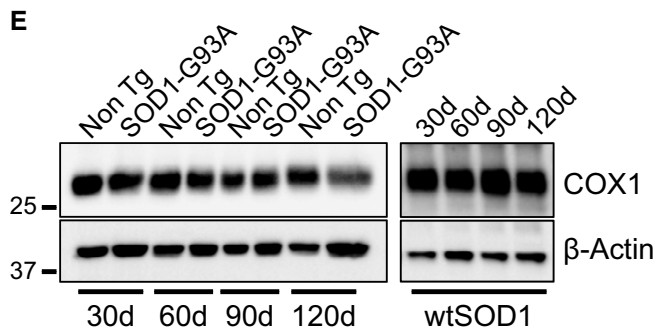

**F**

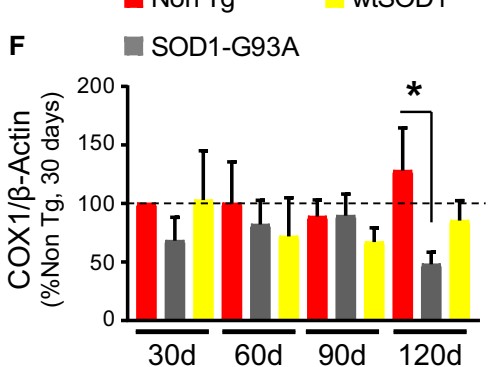

**G**

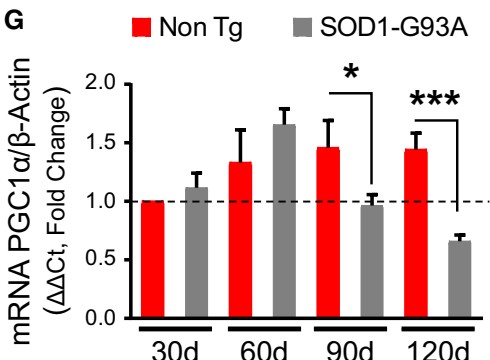

Figure 1.

**Figure 1.  MQC markers increase and mitochondrial content decreases in spinal cord of symptomatic SOD1-G93A mice.**

A, B   Representative Western blot (A) and quantification (B) of p62 associated with mitochondria from spinal cords. Protein levels are normalized by subunit ATPase β of mitochondria (Complex V). Mitochondrial p62 levels are increased at 120 days (symptomatic stage) in SOD1-G93A spinal cord relative to Non Tg and wild-type SOD1 (wtSOD1). Results are expressed as mean ± SEM relative to Non Tg at 30 days; $n = 8$ mice (four males and four females). At 120 days, $**P = 0.0018$ (Non Tg vs. SOD1-G93A) and $**P = 0.0011$ (SOD1-G93A vs. wtSOD1) by paired one-way ANOVA with Tukey's correction. No other statistically significant differences were found (paired Friedman's test with Dunn's correction at 30 days and paired one-way ANOVA with Tukey's correction at 60 and 90 days).

C, D   Representative Western blot (C) and quantification (D) of Tim23 in homogenates from spinal cords using β-actin as normalizer. Results are expressed as mean ± SEM and relative to Non Tg at 30 days; $n = 6$ mice (three males and three females). $***P = 0.0002$ (Non Tg vs. SOD1-G93A at 120 days) by paired one-way ANOVA with Tukey's correction. No other statistically significant differences were found (paired Friedman's test with Dunn's correction at 30 days and paired one-way ANOVA with Tukey's correction at 60 and 90 days). Tim23 is decreased in SOD1-G93A spinal cord relative to Non Tg at 120 days.

E, F   Western blot (E) and quantification (F) of COXI in homogenates from spinal cords. Protein levels are normalized by β-actin. Results are expressed as mean ± SEM and relative to Non Tg at 30 days; $n = 6$ mice (three males and three females). $*P = 0.017$ (Non Tg vs. SOD1-G93A at 120 days) by paired Friedman's test with Dunn's correction. No other statistically significant differences were found (paired Friedman's test with Dunn's correction at 30 and 60 days and paired one-way ANOVA with Tukey's correction at 90 days). COX1 is decreased in SOD1-G93A spinal cord relative to Non Tg at 120 days.

G    qPCR (fold change) of PGC1α mRNA normalized by β-actin mRNA. Results are expressed as mean ± SEM and fold change of Non Tg at 30 days; $n = 6$ (three males and three females) for 30 and 120 days, $n = 4$ (two males and two females) for 60 days and $n = 5$ (three males and two females) for 90 days. $*P = 0.035$ at 90 days and $***P = 0.0007$ at 120 days. Paired Student's $t$-test (for 60, 90, and 120 days) and paired Wilcoxon's test (for 30 days) were used for comparisons.

Source data are available online for this figure.

mitochondrial surface (Narendra *et al*, 2010). Because we found increased mitophagy in SOD1-G93A mice at 60 and 90 days, we then examined Parkin levels in spinal cord homogenates at different time points. We observed a progressive increase in the levels of Parkin in Non Tg mice, as these mice age (Fig 3A and B). Strikingly, the age-related increase in Parkin did not occur in SOD1-G93A. Relative to Non Tg mice, we found significantly less Parkin normalized by β-actin in SOD1-G93A at 90 and 120 days of age (Fig 3A and B). Since by qPCR we did not detect changes in Parkin mRNA in the spinal cord (Fig 3C), data indicate that Parkin decline in SOD1-G93A was not due to transcriptional regulation. We then looked at the levels of Parkin in spinal cord mitochondrial fractions at 120 days of age, when p62 is elevated in SOD1-G93A mice (Fig 1A and B). Parkin levels normalized by Complex V were decreased in SOD1-G93A mice mitochondria relative to Non Tg (Fig 3D and E), suggesting that Parkin recruitment to mitochondria declines at the late disease stage, possibly because the protein is depleted.

**Parkin knockout decreases spinal cord inclusions and recruitment of mitochondrial autophagy markers in SOD1-G93A mice**

The decline in Parkin levels in SOD1-G93A spinal cord mitochondria suggested that the protein becomes depleted because of increased turnover associated with accelerated mitophagy flux. Therefore, we wanted to assess the impact of genetically manipulating Parkin

levels on the disease course. To this end, we crossed SOD1-G93A mice in the C57BL6 background, which have an average lifespan of 157 days, with congenic Parkin knockout mice (Goldberg *et al*, 2003). Parkin knockout (PKO) mice present a very mild phenotype restricted to the nigrostriatal pathway under stress conditions and a lower body weight, with no effects on survival (Palacino *et al*, 2004). We compared the following genotypes, Parkin wild-type/Non Tg (Non Tg), PKO/Non Tg (PKO), Parkin wild-type/SOD1-G93A (G93A), and PKO/SOD1-G93A (PKO/G93A). As expected, PKO mice did not express Parkin in the spinal cord (Fig 4A and B), and Parkin was also decreased (by 53%) in the spinal cord of G93A mice in the C57BL6 background at 130 days, when these mice start developing muscle weakness and decreased body weight. Importantly, Parkin knockout did not affect the content of transgenic mutant SOD1 in the spinal cord at the same age (Fig 4A and C).

As a comparison with spinal cord, we also investigated Parkin levels in two organs typically unaffected by mutant SOD1, cerebellum and liver. At 130 days of age, when Parkin is significantly reduced in spinal cord, we found that cerebellar levels of Parkin were only moderately decreased (20%) in G93A mice (Fig EV1A and B). We also measured p62 levels in cerebellum, which were equally elevated in PKO and PKO/G93A (Fig EV1A and C), indicating that Parkin knockout causes the accumulation of p62 in cerebellum independent of mutant SOD1. In G93A liver at 130 days, there was no significant change in Parkin relative to Non Tg (Fig EV1D

**Figure 2.  Mitophagy is increased in SOD1-G93A spinal cord motor neurons.**

A   Representative images of mt-Keima expressing spinal cord sections of SOD1-G93A and Non Tg mice at 90 days. Mt-Keima fluorescence with excitation at 458 nm is pseudocolored in green and at 543 nm is pseudocolored in red. Emissions were recorded sequentially at 600–650 nm. Scale bar, 150 μm.

B   Representative images of mt-Keima expressing motor neurons in the ventral horn of SOD1-G93A and Non Tg mice at 90 days, imaged and pseudocolored as in (A). Scale bar, 20 μm.

C   Quantification of the rate of mitophagy in motor neurons expressed as the percentage of area with high ratio (543/458 nm) signal normalized by the total mitochondria area. Data were collected from mice at 60 days ($n = 3$, one male and two females), 90 days ($n = 4$, two males and two females), and 120 days ($n = 4$, two males and two females). Number of neurons imaged: 60 days, $n = 77$ Non Tg and 87 SOD1-G93A; 90 days, $n = 118$ Non Tg and 173 SOD1-G93A; and 120 days, $n = 110$ Non Tg and 130 SOD1-G93A. Comparisons within pairs showed that mitophagy is increased in SOD1-G93A spinal cords at 60 and 90 days compared to Non Tg. Results are expressed as mean ± SEM. For 60 days, $**P = 0.0022$; for 90 days, $***P = 0.0001$; and for 120 days, $P = 0.094$, all by unpaired Mann–Whitney test. Comparisons along time showed that mitophagy increases over time in both genotypes. For Non Tg, by unpaired Kruskal–Wallis's test with Dunn's correction, $**P = 0.0046$ and $***P = 0.0001$. For SOD1-G93A, by unpaired Kruskal–Wallis's test with Dunn's correction, $**P = 0.0077$ and $***P = 0.0001$.

Source data are available online for this figure.

   

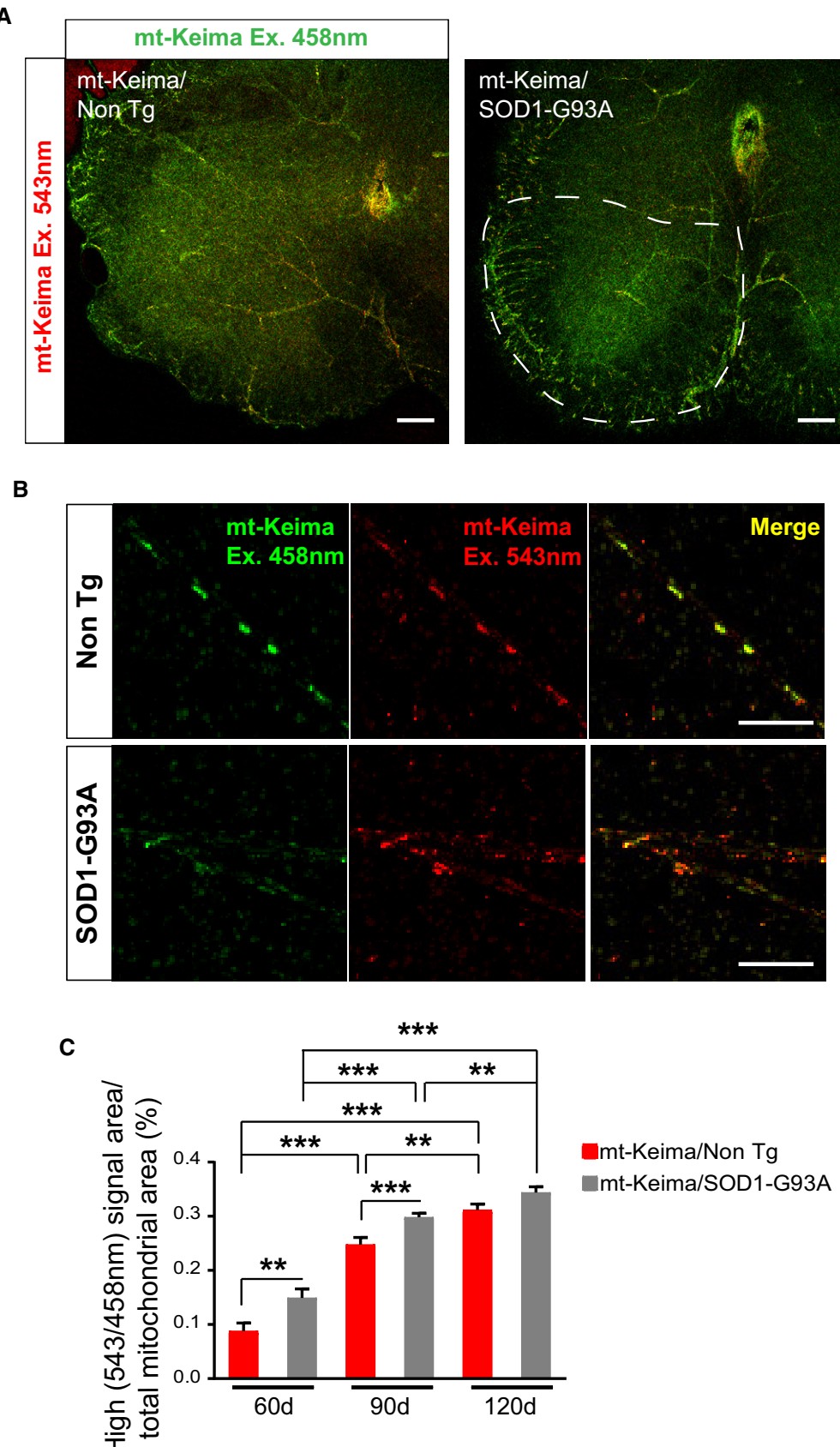

Figure 2.

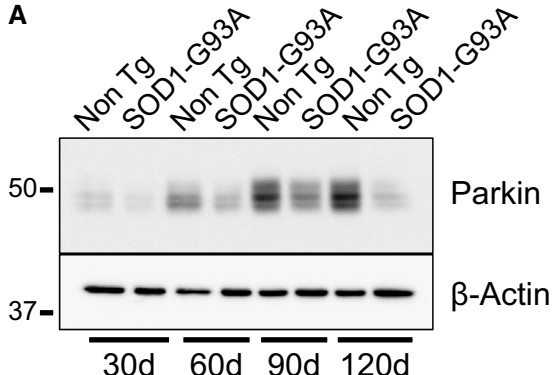

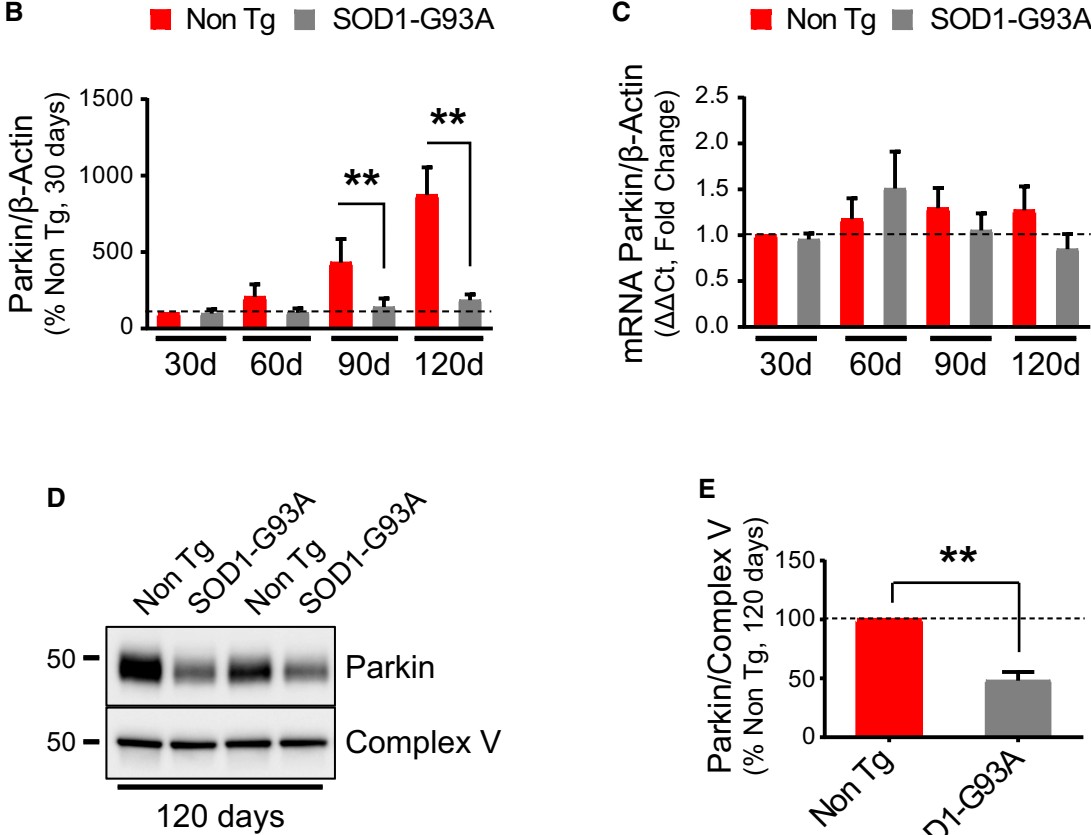

**Figure 3.  Parkin is depleted in the spinal cord of SOD1-G93A mice.**

A, B    Representative Western blot (A) and quantification (B) of Parkin levels in spinal cord homogenates. Protein levels are normalized by β-actin. Results are expressed as mean ± SEM and relative to Non Tg control at 30 days; n = 8 (four males and four females). Paired Wilcoxon's test at 90 days, **P = 0.0078; and paired Student's *t*-test at 120 days, **P = 0.0054. Paired Wilcoxon's test for 30 and 60 days showed no statistically significant differences. Parkin levels are decreased in SOD1-G93A at 90 and 120 days compared to Non Tg.

C    qPCR (fold change) of Parkin mRNA normalized by β-actin mRNA. Results are expressed as mean ± SEM and fold change relative to Non Tg at 30 days of age; n = 6 (three males and three females) for 30 and 120 days, n = 4 (two males and two females) for 60 days, and n = 5 (three males and two females) for 90 days. No statistically significant differences were found by paired Student's *t*-test (90 days) and Wilcoxon's test (30, 60, and 120 days).

D, E    Representative Western blot (D) and quantification (E) of Parkin levels in spinal cord mitochondria at 120 days. Protein levels are normalized by Complex V. Parkin is reduced in spinal cord mitochondria of SOD1-G93A mice at 120 days. Results are expressed as mean ± SEM and shown as percent of Non Tg; n = 8 (four males and four females); **P = 0.0078, by Wilcoxon's *t*-test.

Source data are available online for this figure.

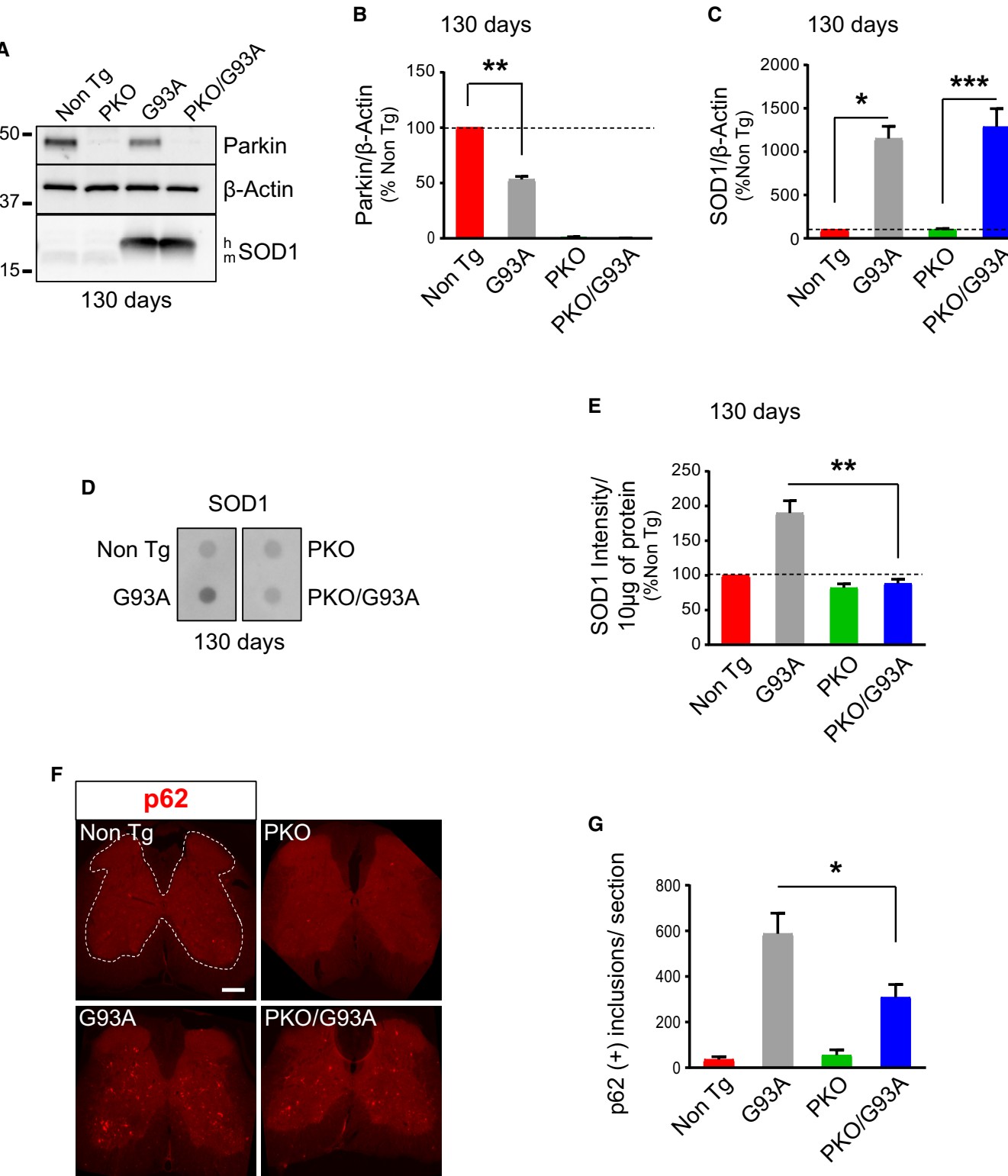

**Figure 4.**

and E). p62 was also unchanged in G93A as well as in PKO and PKO/G93A liver, relative to Non Tg (Fig EV1D and F). These results indicate that Parkin depletion is more pronounced in affected spinal cord than that in unaffected areas of the CNS or in liver.

Since the neuropathology in SOD1-G93A mice involves the formation of large, detergent-insoluble protein aggregates containing SOD1 in the spinal cord (Shaw *et al*, 2008), we assessed the effect of Parkin knockout on these aggregates with a filter-trap

**Figure 4.    p62-positive inclusions in spinal cord of SOD1-G93A mice are decreased by Parkin knockout.**

A    Representative Western blot of spinal cord homogenates at 130 days, showing that Parkin is absent in PKO and PKO/G93A mice.
B    Parkin quantification indicates that Parkin is decreased in G93A mice relative to Non Tg at 130 days. Results are expressed as mean ± SEM and as percent of Non Tg; $n$ = 8 (four males and four females); **$P$ = 0.0078, by Wilcoxon's $t$-test.
C    Quantification of SOD1 at 130 days indicates that Parkin knockout does not affect the levels of transgenic mutant human SOD1 expressed in SOD1-G93A spinal cord. Results are expressed as mean ± SEM and as percent of Non Tg; $n$ = 8 (four males and four females); no statistically significant differences relative to SOD1 expression were found between G93A and PKO/G93A, $P$ = 0.214, by paired Student's $t$-test. Differences between Non Tg and G93A (by paired Friedman's test with Dunn's correction), *$P$ = 0.020 and PKO and PKO/G93A (by paired one-way ANOVA with Tukey's correction), ***$P$ = 0.0001.
D, E    SOD1 filter-trap assay (D) of spinal cord homogenates at 130 days and quantification of dot intensities (E), showing that PKO/G93A mice have less SOD1 aggregates than G93A. Results are expressed as mean ± SEM and as percent of Non Tg; $n$ = 8 (four males and four females); **$P$ = 0.0078 by paired Wilcoxon's test.
F    Representative images of 130-day-old lumbar spinal cord sections immunostained for p62. Scale bar, 200 μm.
G    Quantification of the average number of large p62-positive inclusions per section at 130 days indicates that PKO/G93A mice have less SOD1 aggregates than G93A. Results are expressed as mean ± SEM; $n$ = 7 (three males and four females) mice for Non Tg and PKO, $n$ = 10 (five males and five females) mice for G93A and $n$ = 9 (four males and five females) mice for PKO/G93A; *$P$ = 0.010 by unpaired Mann–Whitney test.

Source data are available online for this figure.

assay. Interestingly, while at 130 days, G93A spinal cords had increased levels of insoluble SOD1 aggregates (Fig 4D and E), PKO/G93A did not. Because it was shown that SOD1 aggregates are formed both in cytosol and in mitochondria of G93A-SOD1 mouse spinal cord (Vijayvergiya *et al*, 2005), we looked at the mitochondrial fractions separately to assess whether they were affected by Parkin knockout. Surprisingly, we did not find differences in SOD1 aggregation between the mitochondrial fractions of G93A and PKO/G93A spinal cords (Fig EV2A and B). This result points to an extra-mitochondrial function of Parkin in regulating the levels of SOD1 aggregation in the cytosol. Alternatively, it is possible that the SOD1 aggregates affected by Parkin only form on the surface of the outer membrane of mitochondria and are lost upon mitochondrial isolation.

Next, we performed immunostaining of lumbar spinal cord sections at 130 days of age. Numerous large p62 aggregates were detectable in the gray matter of G93A spinal cords (Fig 4F), but p62-positive inclusions were significantly less abundant in PKO/G93A mice (Fig 4G). High-magnification images showed p62 aggregates in choline acetyltransferase (ChAT)-positive motor neurons of G93A mice and PKO/G93A mice, both in the soma and the neuronal processes, but motor neurons containing p62 inclusions were significantly less in PKO/G93A (Fig 5A and B). Taken together, these results suggest that loss of Parkin either prevents the formation of intracellular inclusions tagged for autophagy or increases their clearance.

Then, we assessed the effects of Parkin knockout on mitophagy markers in the spinal cord mitochondrial fractions at 130 days. We first looked at the autophagy receptor optineurin (OPTN). Notably, mutations in OPTN have been associated with familial motor neuron disease (Maruyama *et al*, 2010). The mitochondrial content of OPTN normalized by Complex V was lower in PKO/G93A than G93A mice (Fig 5C and D). Mitochondrial p62 was also significantly less in PKO/G93A compared to G93A (Fig 5C and E). We note that the increase in p62 in G93A mice at this age is more pronounced than that OPTN, suggesting that in the spinal cord exposed to mutant SOD1, at least at this disease stage, the MQC response is more dependent on p62 than OPTN. Additionally, we determined the content of VCP, which extracts ubiquitinated proteins from the OMM and delivers them to the proteasome, allowing mitophagy to progress (Tanaka *et al*, 2010). The amount of VCP in spinal cord mitochondrial fractions was significantly less in PKO/G93A than G93A (Fig 5C and F). At disease end stage, the mitochondrial levels of p62, OPTN, and VCP were markedly increased by several folds, in both G93A and PKO/G93A relative to Non Tg and PKO, respectively (Fig EV3A–E). While the p62 levels at end stage did no longer significantly differ between G93A and PKO/G93A, OPTN and VCP were still lower in PKO/G93A.

Taken together, these results indicate that Parkin modulates the recruitment of MQC proteins to mitochondria in G93A mice, since in PKO/G93A mice, the abundance of MQC markers associated with spinal cord mitochondria was overall decreased compared to G93A.

**Figure 5.    The content of mitochondrial p62, OPTN, and VCP in spinal cord of SOD1-G93A mice is decreased by Parkin knockout.**

A    Immunostaining of anterior horn of lumbar spinal cord for ChAT (in green) and p62 (in red) at 130 days. Scale bar, 10 μm.
B    Quantification of the percentage of ChAT-positive motor neurons (MN) containing p62-positive inclusions at 130 days. PKO/G93A MN have less SOD1 aggregates than G93A, while Non Tg and PKO MN have none. Data were collected from $n$ = 4 (two males and two females) mice for Non Tg and PKO and $n$ = 6 (three males and three females) for G93A and PKO/G93A. Number of images recorded for each genotype: $n$ = 9 for Non Tg and PKO, $n$ = 16 for G93A, and $n$ = 12 for PKO/G93A. Results are expressed as mean ± SEM; *$P$ = 0.032, by unpaired one-tailed Student's $t$-test.
C    Representative Western blots of OPTN, p62, and VCP in spinal cord mitochondrial fractions at 130 days. Protein levels are normalized by Complex V. The quantifications in panels (D–F) show that mitochondria of PKO/G93A mice have less mitophagy adaptor proteins than G93A relative to Complex V.
D    Quantification of OPTN relative to Complex V at 130 days. Results are expressed as mean ± SEM and as percent of Non Tg; $n$ = 8 (four males and four females); *$P$ = 0.039 by paired Student's $t$-test.
E    Quantification of p62, relative to Complex V at 130 days. Results are expressed as mean ± SEM and as percent of Non Tg; $n$ = 8 (four males and four females); **$P$ = 0.0029 by paired Student's $t$-test.
F    VCP was quantified using Complex V as loading reference at 130 days. Results are expressed as mean ± SEM and as percent of Non Tg; $n$ = 8 (four males and four females); *$P$ = 0.044 by paired Student's $t$-test.

Source data are available online for this figure.

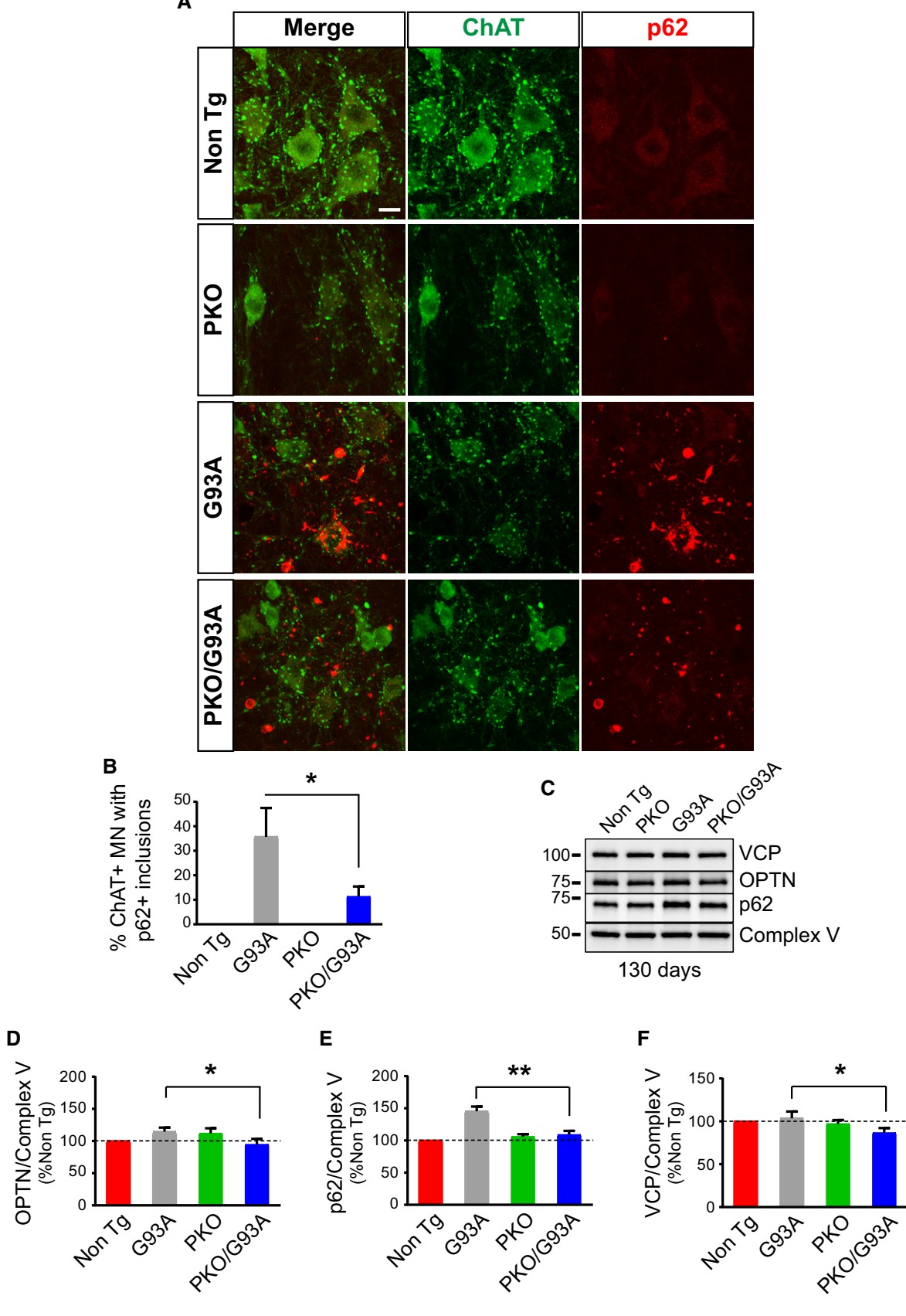

**Figure 5.**

### Parkin knockout is a disease modifier in SOD1-G93A mice

To evaluate the impact of Parkin ablation on the disease phenotype in the SOD1-G93A mice, we assessed their lifespan. We observed a significant increase in survival in the PKO/G93A compared to G93A mice (189.3 ± 2.5 days compared to 169.1 ± 2.2 days; $P = 0.00003$; Fig 6A). G93A and PKO/G93A mice showed a decline in body weight starting at 130 days, and this decline was slower in PKO/G93A compared to G93A mice (Fig 6B). Another, earlier sign of neurodegeneration in mice is hind limb clasping, which occurred at the same age (94–101 days) in G93A and PKO/G93A mice (Fig 6C). Together, these data indicate that while disease onset is unmodified by Parkin ablation, disease progression is delayed in the PKO/G93A relative to G93A mice.

To better understand the effects of Parkin knockout on the pathology of G93A mice, we performed histological studies at 130 days. First, since muscle denervation is a key pathological event that causes paralysis in ALS, we examined the neuromuscular junctions (NMJ) in the soleus muscle. We found a severe decrease in the proportion of innervated NMJ in 130-day-old G93A mice, which were significantly attenuated in PKO/G93A mice (Fig 6D and E). The timing of NMJ denervation in G93A mice has been previously reported (Fischer *et al*, 2004). It starts around 50 days of age and progresses throughout the course of the disease. Therefore, the decline in innervated NMJs occurs in parallel with the decrease in Parkin levels, and it is delayed by Parkin knockout.

Next, we estimated the number of ChAT-positive motor neuron somas in the anterior horn of the lumbar spinal cord. There was a significant decline in ChAT-positive neurons in G93A, but not in PKO/G93A mice (Fig 6F and G). Lastly, we looked at glial fibrillary acidic protein (GFAP) staining in lumbar spinal cord, which is characteristically increased in ALS and is indicative of astrogliosis (Schiffer *et al*, 1996). PKO/G93A spinal cord had less intense GFAP staining compared to G93A, indicating a decrease in astrogliosis (Fig 6H and I).

Taken together, phenotypic and pathologic studies on PKO/G93A mice indicate that Parkin is a disease modulator in SOD1-G93A mice and that the ablation of Parkin results in an attenuation of pathology and a delay in disease progression.

### Parkin knockout delays mitochondrial decline in SOD1-G93A spinal cord

Increased rate of mitochondrial turnover accompanied by impaired mitochondrial biogenesis in SOD1-G93A spinal cord could lead to a decline in the pool of functional mitochondria. Increased mitochondrial turnover in the double transgenic mice was indicated by the decrease in Tim23 and COXI protein levels (Fig EV4). Thus, to assess mitochondrial function, we measured the activities of the Krebs cycle enzyme citrate synthase and the respiratory chain Complex IV (COX), which is typically decreased in SOD1-G93A spinal cords (Mattiazzi *et al*, 2002), in spinal cord homogenates at disease end stage. Although a reduction in both enzymatic activities was detected in G93A and PKO/G93A mice, citrate synthase (Fig 7A) and COX (Fig 7B) activities were significantly higher in PKO/G93A compared to G93A mice. These results suggest that Parkin ablation attenuates the decline in mitochondrial enzyme activities in SOD1-G93A mice. G93A mice also had reduced levels of

PGC1α mRNA at 130 days (Fig 7C), which declined further at end stage (Fig 7D). Interestingly, no decrease in PGC1α mRNA was detected in PKO/G93A mice at 130 days, while at end stage, PGC1α mRNA declined to levels similar to those of G93A mice. The ablation of Parkin did not affect PGC1α expression in the spinal cord, as PGC1α mRNA did not differ between Non Tg and PKO mice. Taken together, these results suggest that impaired mitochondrial biogenesis in the spinal cord through downregulation of PGC1α is delayed by Parkin knockout.

Parkin was shown to interact with the cytosolic protein PARIS (ZNF746) and facilitate its degradation (Shin *et al*, 2011). When PARIS is stabilized, such as in Parkin knockout models, it can interact with PGC1α and accelerate its turnover. Since PGC1α expression was decreased in G93A mice, we measured PARIS protein levels at disease end stage (Fig 7E and F). PARIS was unchanged in G93A and PKO/G93A, suggesting that loss of Parkin did not increase PARIS levels in the spinal cord, and therefore, PGC1α decline was not due to increased PARIS.

### Parkin knockout delays the decline of mitochondrial dynamics proteins in SOD1-G93A spinal cord

It was proposed that mitophagy does not progress unless proteins of the OMM involved in mitochondrial dynamics are degraded via the proteasome (Tanaka *et al*, 2010). Therefore, we assessed the levels of known OMM mitochondrial dynamics proteins ubiquitinated by Parkin, namely Miro1, a protein involved in the regulation of mitochondrial transport, and mitofusin 2 (Mfn2), involved in OMM fusion. Levels of both Miro1 (Fig 8A and B) and Mfn2 (Fig 8C and D) normalized by β-actin were decreased in spinal cord homogenates from end stage G93A and PKO/G93A mice, but significantly higher in PKO/G93A than in G93A, suggesting that Parkin knockout delays the turnover of OMM proteins involved in mitochondrial dynamics.

p62 accumulation in spinal cord mitochondria of SOD1-G93A mice was attenuated by Parkin knockout and the levels of Mfn2 and Mito1 preserved, but these effects were partial and the disease progression was delayed but not arrested, suggesting that the levels of other E3-ubiquitin ligases could be normally maintained in G93A mitochondria throughout disease course, even in the absence of Parkin. To test this hypothesis, we investigated protein ubiquitination profiles in spinal cord mitochondrial fractions, using antibodies that recognize ubiquitin chains K48 or K63. The former tags proteins for proteasomal degradation and the latter for autophagy (Narendra *et al*, 2010). Differences in mitochondrial ubiquitinated proteins were apparent at end stage with both antibodies (Fig 8E). These differences, however, correlated with the presence or absence of the SOD1-G93A transgene, rather than with Parkin, as the profiles of ubiquitinated proteins differed between G93A and Non Tg and between PKO and PKO/G93A, but not between G93A and PKO/G93A.

Mitochondrial protein ubiquitination leads to recruitment of autophagosome machinery components, starting with LC3 II. The levels of LC3 II in spinal cord mitochondria at disease end stage were similarly increased in both G93A and PKO/G93A (Fig 8F and G). Taken together, these data suggested that at end-stage G93A and PKO/G93A spinal cords have similar MQC activities.

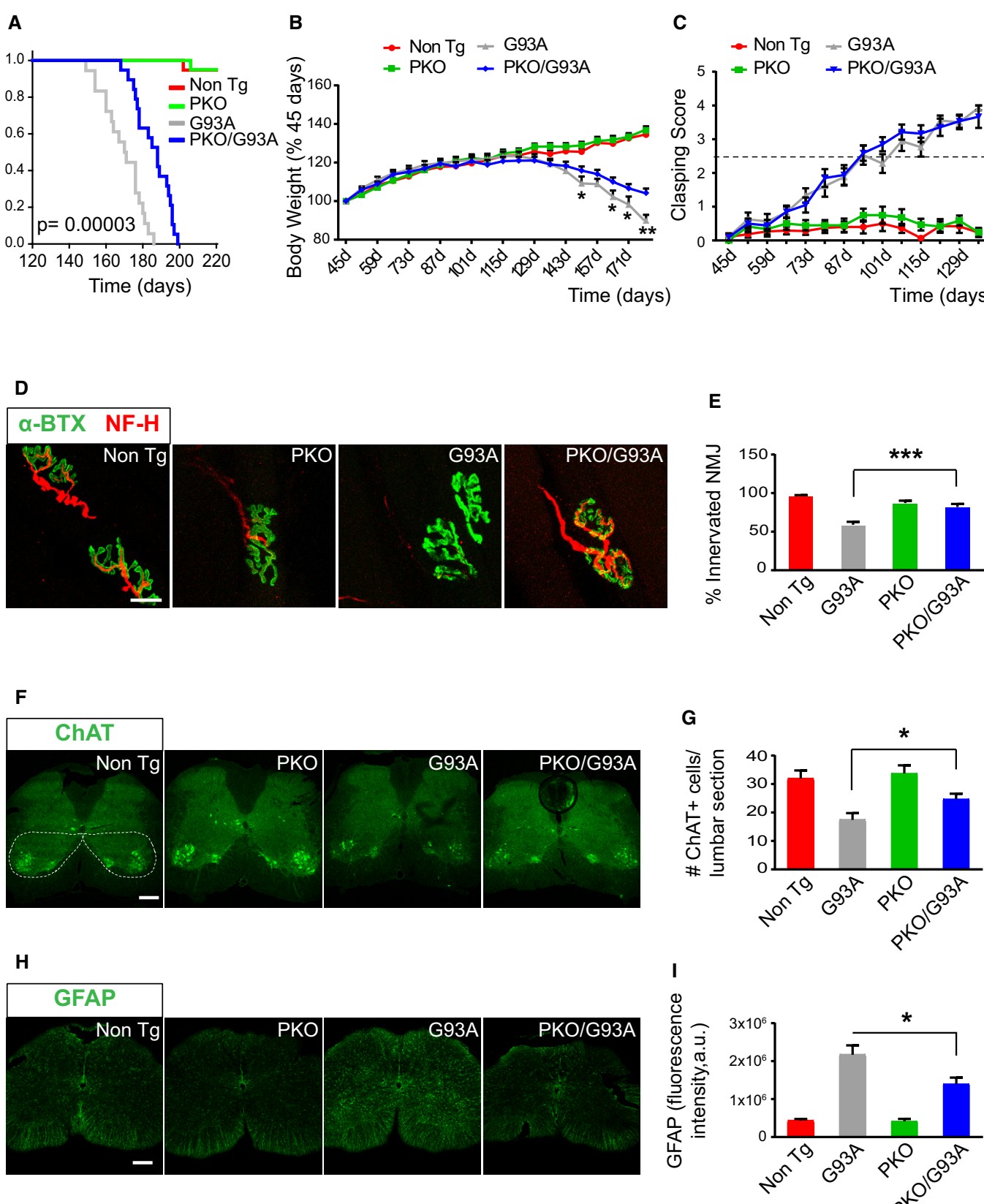

**Figure 6.**

**Figure 6. Parkin knockout delays disease progression and pathology in SOD1-G93A mice.**

A  Kaplan–Meier survival analysis (log-rank test with Holm–Sidak's correction for multiple comparisons). Mean age of death ($\pm$SEM) for G93A was 169.3 $\pm$ 2.5 days and for PKO/G93A was 185.6 $\pm$ 2.2 days, indicating that Parkin knockout prolongs lifespan of SOD1-G93A mice; $n$ = 19 for Non Tg (nine males and 10 females), $n$ = 20 for PKO (10 males and 10 females), $n$ = 18 for G93A (eight males and 10 females), and $n$ = 19 for PKO/G93A (nine males and 10 females); $P$ = 0.0000285.

B  Body weight changes over time, relative to body weight at 45 days that is set at 100% for each mouse. Parkin knockout delays body weight loss in SOD1-G93A mice. Results are expressed as mean $\pm$ SEM; $n$ = 19 for Non Tg (nine males and 10 females), $n$ = 20 for PKO (10 males and 10 females), $n$ = 18 for G93A (eight males and 10 females), and $n$ = 19 for PKO/G93A (nine males and 10 females); *$P$ = 0.017 (at 150 days), *$P$ = 0.004 (at 164 days), *$P$ = 0.002 (at 171 days), and **$P$ = 0.001 (at 178 days), by two-way ANOVA with repeated measures and with Holm–Sidak's correction for multiple comparisons.

C  Clasping score changes over time. Parkin knockout does not affect clasping onset in SOD1-G93A mice. An intermediate score of 2.5 is indicated by the dotted line. Results are expressed as mean $\pm$ SEM; $n$ = 19 for Non Tg (nine males and 10 females), $n$ = 20 for PKO (10 males and 10 females), and $n$ = 18 for G93A (eight males and 10 females), and $n$ = 19 for PKO/G93A (nine males and 10 females).

D  Representative images of NMJs in the soleus muscle at 130 days. Muscles were immunostained for neurofilament H (NF-H, in red) and stained with $\alpha$-bungarotoxin ($\alpha$-BTX, in green). Scale bar, 20 $\mu$m.

E  Quantification of the percentage of innervated NMJs at 130 days indicates that Parkin knockout decreases denervation in SOD1-G93A mice. Results are expressed as mean $\pm$ SEM and were collected from $n$ = 6 mice per group (three males and three females); $n$ of images evaluated = 49 for Non Tg (with 483 NMJs counted), 46 for PKO (396 NMJs counted), 47 for G93A (413 NMJs assessed), and 38 for PKO/G93A (325 NMJs counted); ***$P$ = 0.0001, by unpaired Mann–Whitney test.

F  Representative images of lumbar spinal cord sections at 130 days immunostained with ChAT. Scale bar, 200 $\mu$m.

G  Quantification of ChAT-positive MN in the spinal cord ventral horn (dotted areas) at 130 days shows that Parkin knockout decreases MN loss in the spinal cord of SOD1-G93A mice. Results are expressed as mean $\pm$ SEM; $n$ = 10 mice per group in Non Tg, PKO, and G93A (five males and five females) and $n$ = 9 for PKO/G93A (four males and five females); *$P$ = 0.031, by unpaired Student's $t$-test.

H  Representative images of lumbar spinal cord sections immunostained for GFAP at 130 days. Scale bar, 200 $\mu$m.

I  Quantification of the average GFAP staining intensity at 130 days indicates that Parkin knockout decreases spinal cord astrogliosis in SOD1-G93A mice. Results are expressed as mean $\pm$ SEM; $n$ = 10 mice per group (five males and five females); *$P$ = 0.033, by paired Student's $t$-test.

Source data are available online for this figure.

March5 is a mitochondria-resident E3-ubiquitin ligase, which has been reported to interact with and to ubiquitinate Mfn2 (Yonashiro *et al*, 2006; Karbowski *et al*, 2007; Park *et al*, 2010). March5 was also found to be involved in mutant SOD1 degradation (Yonashiro *et al*, 2009). Therefore, we analyzed the mitochondrial content of March5 at end stage and found that its levels normalized by Complex V were unchanged in G93A, PKO, and PKO/G93A spinal cords relative to Non Tg (Fig 8H and I). In addition, we analyzed the levels of another mitochondrial resident E3-ubiquitin ligase, Mulan (MUL1), which is also capable of triggering Mfn2 degradation and mitophagy (Cilenti *et al*, 2014; Rojansky *et al*, 2016). Levels of MUL1 were also unchanged in G93A, PKO, and PKO/G93A spinal cords relative to Non Tg (Fig 8H and J). These findings indicate that, unlike Parkin, the resident mitochondrial E3-ubiquitin ligases March5 and MUL1 were stable in spinal cord mitochondria of G93A mice. Since the targets of these E3-ubiquitin ligases overlap, and include mitochondrial dynamics proteins (Covill-Cooke *et al*, 2017), such as Mfn2, it is not surprising that the ablation of Parkin can only exert a partial protective effect on SOD1-G93A mitochondria.

**Parkin overexpression worsens cell toxicity induced by mitochondrial selective oxidative stress damage in motor neuron-like NSC34 cells**

Based on the finding that Parkin knockout delays disease progression in mutant SOD1 mice, we postulated that Parkin expression could worsen phenotypes associated with mitochondrial damage in motor neurons. To start testing this hypothesis we took advantage of NSC34 motor neuron-like cells stably transfected with human G93A mutant SOD1 (Magrane *et al*, 2009) or mock transfected with vector control. Notably, NSC34 cells do not express endogenous Parkin (Fig EV5A). These cells were transiently transfected with vectors expressing either YFP-Parkin or pEGFP-N1. Cells were then exposed to mitochondrial-specific oxidative stress with mitoPQ, a form of paraquat selectively targeted to mitochondria (Robb *et al*,

2015), and assessed YFP-Parkin intracellular distribution and the viability of transfected cells. While GFP showed the expected diffuse fluorescence, upon mitoPQ treatment YFP-Parkin clustered with mitochondria, reflecting its recruitment to damaged mitochondria (Fig EV5B). Furthermore, mitoPQ caused cell death only in cells expressing YFP-Parkin and not in cells expressing GFP, and the toxicity was more pronounced in SOD1-G93A cells expressing YFP-Parkin (Fig EV5C). Together, these results support the hypothesis that Parkin exacerbates the effects of mitochondrial damage.

## Discussion

Here, we show that mutant SOD1-G93A activates mitophagy in the spinal cord of ALS mice. Mitophagy in SOD1-G93A spinal cord was assessed in living motor neurons of fresh spinal cord sections using mt-Keima (Sun *et al*, 2015). To our knowledge, this is the first demonstration of the use of this reporter to investigate mitophagy in a mouse model of neurodegeneration. We also detect accumulation of mitochondrially localized MQC proteins, p62 and OPTN, suggesting that ultimately the ability to eliminate damaged mitochondria through mitophagy in SOD1-G93A spinal cord is compromised. Similarly, the levels of general autophagy markers, such as LC3 II, were strongly increased in the late disease stages, indicating that the autophagy program is active, but degradation is stalling. These findings are in accord with altered late endosome transport defects and autophagosome degradation impairment in mutant SOD1 mice (Xie *et al*, 2015).

We find an unexpected decline in Parkin levels, in SOD1-G93A spinal cords relative to Non Tg controls, while Parkin mRNA was unchanged throughout disease course, suggesting that transcriptional regulation is not involved in Parkin decline. It is also noteworthy that the total levels of Parkin protein in spinal cord increased with age in control mice, but not in mutant SOD1 mice. Together, these results suggest that Parkin depletion is due to

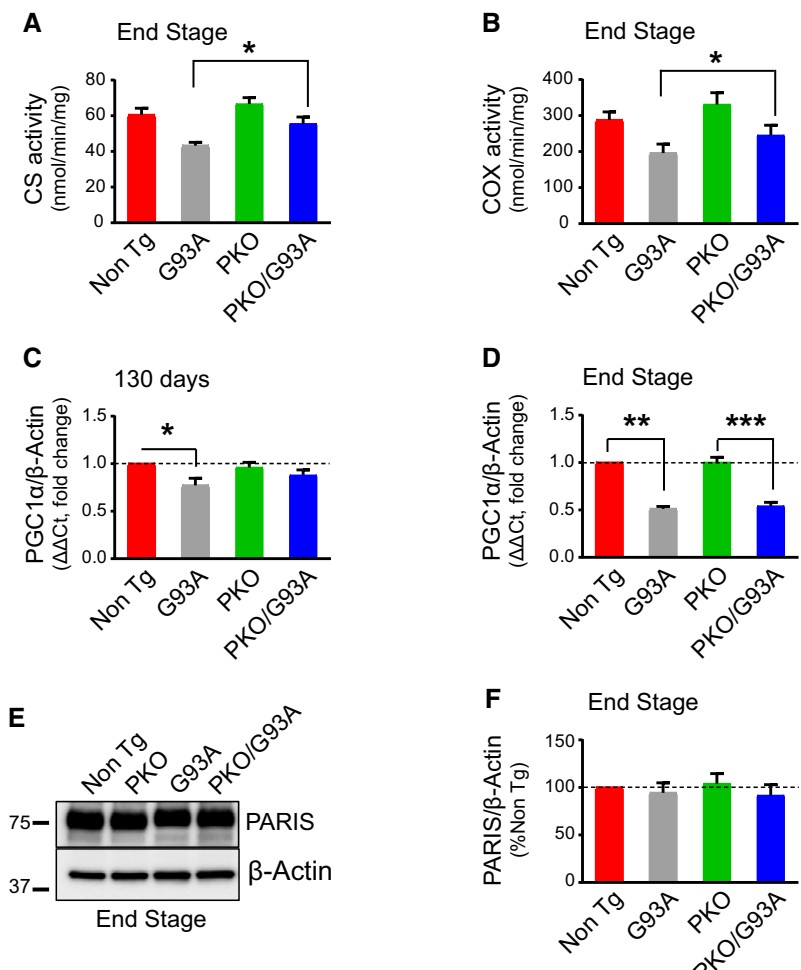

**Figure 7. Parkin knockout improves mitochondrial dysfunction and delays PGC1α decline in SOD1-G93A mice.**

A    Citrate synthase activity in spinal cord homogenates at disease end stage. Results are expressed as mean ± SEM; *n* = 8 (four males and four females) mice per group; *P = 0.024, by paired Student's *t*-test. Activities are expressed as nmoles of substrate oxidized per minute per mg of protein.

B    Cytochrome oxidase (COX) activity at end stage. Results are expressed as mean ± SEM; *n* = 8 (four males and four females) mice per group; *P = 0.032, by paired Student's *t*-test. Activities are expressed as nmoles of substrate oxidized per minute per mg of protein.

C    qPCR (fold change) of PGC1α mRNA normalized by β-actin mRNA at 130 days of age. Results are expressed as mean ± SEM and as fold change of the Non Tg; *n* = 8 (four males and four females). No statistically significant differences were found between G93A and PKO/G93A (*P* = 0.250 by paired Wilcoxon's test); *P = 0.035 paired Friedman's test with Dunn's correction (Non Tg vs. G93A). No other statistically significant differences were found.

D    qPCR (fold change) of PGC1α mRNA normalized by β-actin mRNA at end stage. Results are expressed as mean ± SEM and as fold change of the Non Tg; *n* = 8 (four males and four females). No statistically significant differences were found between G93A and PKO/G93A (*P* = 0.613 by paired Student's *t*-test). **P = 0.0058 by paired Friedman's test with Dunn's correction (for Non Tg vs. G93A comparison) and ***P < 0.0001 by paired one-way ANOVA with Tukey's correction (for PKO vs. PKO/G93A comparison). Parkin knockout improves PGC1α expression at 130 days (C) but the effect is lost at end stage (D).

E, F  Representative Western blots (E) and quantification (F) of PARIS at disease end stage. Protein levels are normalized by β-actin, and results are expressed as mean ± SEM and as percent of Non Tg; *n* = 8 (four males and four females). No statistically significant differences were found between G93A and PKO/G93A by paired Wilcoxon's test (*P* = 0.99). PARIS levels are unchanged in SOD1-G93A mice relative to Non Tg at disease end stage.

Source data are available online for this figure.

increased protein turnover, as a result of enhanced MQC, which combined with decreased PGC1α levels cause a decline in the pool of mitochondrial protein.

The finding of Parkin depletion was most pronounced in spinal cord, the tissue most severely affected in SOD1-G93A mice, prompted us to ask whether modulating Parkin levels modifies the pathology and the disease phenotype. We utilized the constitutive PKO mouse, because it does not present overt neurodegenerative phenotypes (Goldberg *et al*, 2003; Kitada *et al*, 2009), possibly due

to compensatory mechanisms (Dawson *et al*, 2010). We reasoned that although the PKO mouse may be compensated under normal circumstances, in the presence of extensive and chronic mitochondrial damage induced by mutant SOD1 (Mattiazzi *et al*, 2002; Damiano *et al*, 2006), Parkin knockout may result in an accelerated disease course, due to defective MQC. A similar hypothesis was recently explored using the "mutator" mouse (i.e., a mouse that accumulates mtDNA mutations), in which constitutive Parkin knockout induced a neurodegenerative phenotype (Pickrell *et al*,

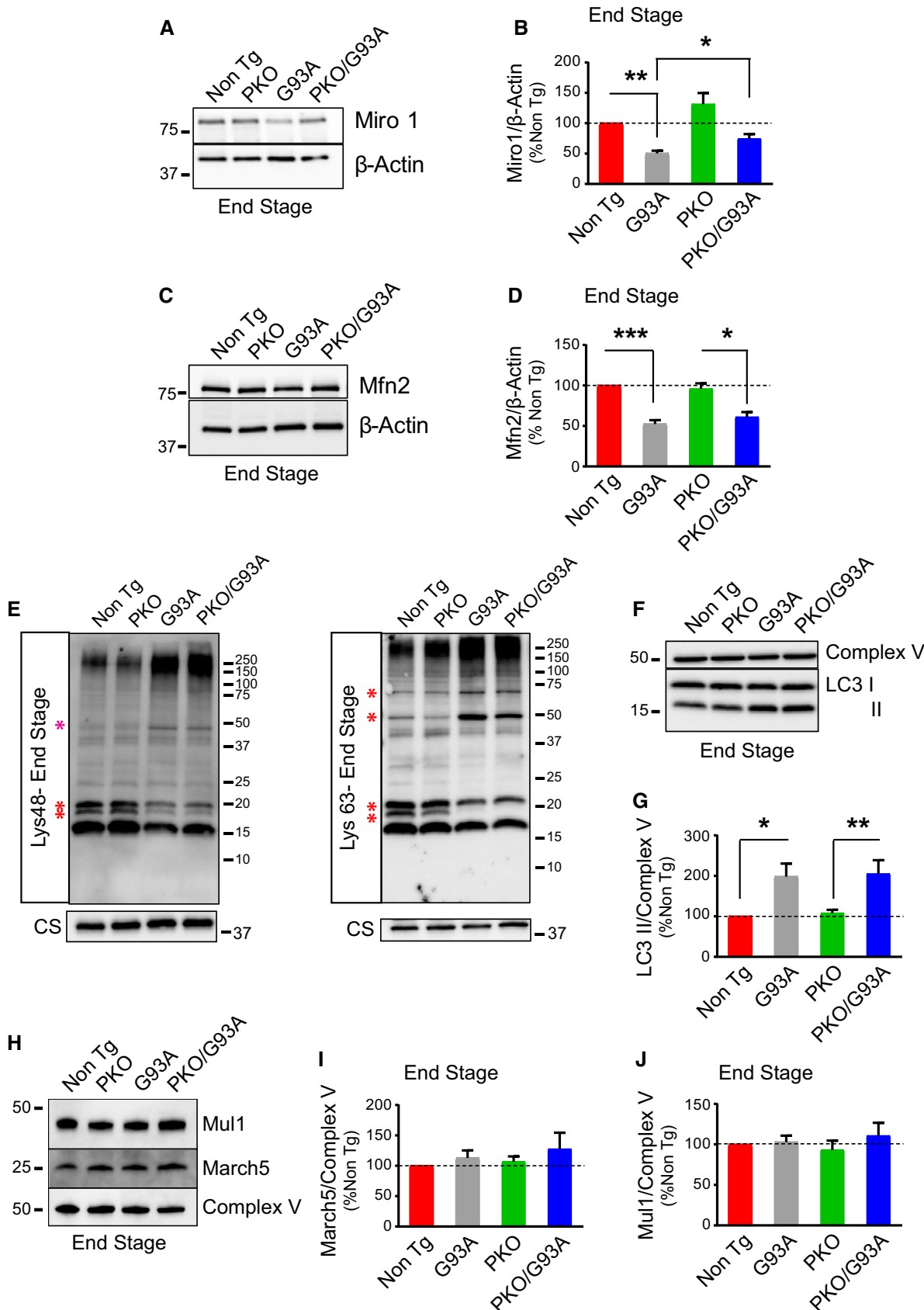

Figure 8.

◄

**Figure 8.   Parkin knockout improves the decline of mitochondrial dynamics proteins in SOD1-G93A spinal cord, but mitochondrial protein ubiquitination profiles and March5 and Mul1 levels are unaffected.**

A, B    Western blots (A) and quantification (B) of Miro1 in spinal cord homogenates at disease end stage. β-actin is used as loading reference. Results are expressed as mean ± SEM and as percent of Non Tg; $n = 8$ (four males and four females) mice per group; *$P = 0.011$ by paired Student's $t$-test (for comparison G93A vs. PKO/G93A) and **$P = 0.003$ by paired Friedman's test with Dunn's correction (for Non Tg vs. G93A). Parkin knockout increases the levels of Miro1 in SOD1-G93A mice at disease end stage.

C, D    Western blots (C) and quantification (D) of Mfn2 in spinal cord homogenates at disease end stage. Protein levels are normalized by β-actin. Results are expressed as mean ± SEM and as percent of Non Tg; $n = 8$ (four males and four females) mice per group. No statistically significant differences were found between G93A and PKO/G93A ($P = 0.078$ by paired Wilcoxon's test); ***$P = 0.0007$ by paired Friedman's test with Dunn's correction (for Non Tg vs. G93A) and *$P = 0.037$ by paired Friedman's test with Dunn's correction (for PKO vs. PKO/G93A). Parkin knockout increases the levels of Mfn2 in SOD1-G93A mice at disease end stage.

E    Western blots of spinal cord mitochondria at disease end stage probed for lysine 48 (left panel) and lysine 63 (right panel) ubiquitin chains. Citrate synthase is used as loading control. Asterisks indicate ubiquitinated proteins with different abundance in SOD1-G93A samples.

F, G    Representative Western blot (F) and quantification (G) of LC3 II in spinal cord mitochondria at end stage. Protein levels are normalized by Complex V. LC3 II is increased in both G93A and PKO/G93A mitochondria and Parkin knockout does not affect LC3 II levels. Results are expressed as mean ± SEM and as percent of Non Tg; $n = 8$ (four males and four females) mice per group; No statistically significant differences were found between G93A and PKO/G93A ($P = 0.504$ by paired Student's $t$-test). *$P = 0.035$ by paired Friedman's test with Dunn's correction (for Non Tg vs. G93A) and **$P = 0.0014$ by paired one-way ANOVA with Tukey's correction (for PKO vs. PKO/G93A).

H    Representative Western blot of March5 and Mul1 in spinal cord mitochondria at end stage.

I    Quantification of March5 at disease end stage. Results are expressed as mean ± SEM and percent of Non Tg; $n = 8$ (four males and four females) mice per group. No statistically significant differences were found between G93A and PKO/G93A by paired Wilcoxon's test ($P = 0.742$). Protein levels are normalized by Complex V. March5 levels are unaffected by Parkin knockout.

J    Quantification of Mul1 at disease end stage. Results are expressed as mean ± SEM and percent of Non Tg; $n = 8$ (four males and four females) mice per group. No statistically significant differences were found between G93A and PKO/G93A by paired Wilcoxon's test ($P = 0.94$). Protein levels are normalized by Complex V. Mul1 levels are unaffected by Parkin knockout.

Source data are available online for this figure.

2015). To our surprise, Parkin knockout resulted in a delay in body weight loss and a substantial survival extension in SOD1-G93A mice. This phenotypic improvement correlated with delayed motor neuron loss, astrogliosis, and NMJ degeneration.

At the histopathological level, a lower concentration of p62-positive inclusions was found in the spinal cord of PKO/G93A mice, likely due to decreased mutant SOD1 aggregation. Surprisingly, the decrease in aggregates did not involve the mitochondrial fractions. These findings suggest accumulation of Parkin-mediated ubiquitination of mutant SOD1 in the cytosol or peripherally associated with the mitochondrial OMM. This accumulation could provide the seed for the formation of the large insoluble SOD1 aggregates, resulting in large p62/ubiquitin-positive intracellular inclusions commonly found in ALS motor neurons (Gal *et al*, 2007).

The different outcomes of Parkin knockout in the "mutator" (i.e., phenotypic worsening) and the SOD1-G93A mice may be due to the differences in the topology of protein misfolding. In the former, mutant mtDNA-encoded proteins accumulate in the matrix, while in the latter they come from the cytosol and accumulate in the OMM and IMS. Therefore, these different localizations could trigger different MQC responses involving different sets of enzymes and signaling pathways.

The finding that Parkin knockout improves the phenotype of SOD1-G93A mice was unexpected, but not completely unprecedented, since Pink1 and Parkin were identified as modifiers of mutant FUS, and Parkin knockout improved the phenotype of flies overexpressing human FUS (Chen *et al*, 2016). Furthermore, Parkin knockout in cardiac myocytes improved the cardiomyopathy induced by concomitant Drp1 KO, as well as decreased mitochondrial protein ubiquitination and association of p62 with mitochondria (Song *et al*, 2015). This evidence combined with our result in a genetic mouse model of ALS, suggests that Parkin knockout can have beneficial effects under certain types of chronic mitochondrial damage. In particular, protracted activation of MQC in response to accumulation

of misfolded proteins on the mitochondrial OMM and in the IMS, as in the case of mutant SOD1 (Vijayvergiya *et al*, 2005; Vande Velde *et al*, 2008; Pickles *et al*, 2016), could trigger accelerated turnover of OMM proteins, either as innocent bystanders or because they also become misfolded. These proteins could be key players of mitochondrial dynamics, such as Mfn2 and Miro1, which are known targets of ubiquitination and degradation by Parkin and other Ub ligases, such as March5 and MUL1 (Karbowski & Youle, 2011). In addition to triggering mitophagy through the PINK/Parkin "canonical" pathway, OMM protein ubiquitination can also lead to their degradation by the proteasome, after being extracted from the membrane by VCP (Xu *et al*, 2011). This pathway could be particularly relevant to our study, as VCP recruitment to mitochondria requires ubiquitination of mitochondrial targets (Kim *et al*, 2013), and we find that VCP is increasingly accumulated on mitochondria of SOD1-G93A mice. Lastly, Parkin also intervenes in mitochondria-derived vesicles, a form of MQC that removes only portions of damaged mitochondria (Sugiura *et al*, 2014), and has not yet been investigated in the central nervous system.

Our results suggest that chronically accelerated turnover of mitochondrial OMM proteins involved in mitochondrial dynamics and transport leads to their depletion resulting in the defects of mitochondria fusion and transport, which are well characterized in mutant SOD1 animal and cellular models and in other forms of ALS, such as TDP-43 mutants (Magrane & Manfredi, 2009; Smith *et al*, 2017). For example, Parkin-mediated degradation of Miro1 blocks mitochondrial axonal transport and facilitates mitophagy in neurons (Wang *et al*, 2011; Liu *et al*, 2012; Birsa *et al*, 2014). Therefore, chronically increased OMM proteostasis and mitophagy in spinal cord of mutant SOD1 can lead to mitochondrial mislocalization and depletion. Increased mitophagy in SOD1-G93A mice was indicated by mt-Keima and by the mitochondrial accumulation of p62 and the neuronal mitophagy receptor OPTN (Wong & Holzbaur, 2014).

Parkin knockout delays and attenuates the removal of OMM proteins and the recruitment of mitophagy receptors. However, this effect is only partial and limited in time. The onset of clasping and body weight loss was not affected in the PKO/G93A relative to G93A mice, but the progression to death was delayed by more than 20 days. Therefore, when we assessed the biochemical phenotype in the spinal cord of end-stage mice, the PKO/G93A mice had progressed for 20 days longer than the G93A littermates, at which point most of the beneficial effects of Parkin knockout had been lost. The most probable explanation for the limitations of Parkin knockout in modifying the MQC process in the spinal cord of SOD1-G93A mice lies on the large overlap in the ubiquitination function with other ligases. Of the various mitochondrial ubiquitin ligases (Covill-Cooke *et al*, 2017), we studied March5 and MUL1, because they are known to be expressed in the nervous system, but it is possible that other ubiquitin ligases are involved. In support of this interpretation, it was recently demonstrated that in transgenic mice expressing a mitophagy reporter, the ablation of PINK1 did not affect the rate of basal mitophagy in neurons (McWilliams *et al*, 2018), further indicating that the loss of PINK1-Parkin in the nervous system can be complemented by other MQC mechanisms.

Parkin knockout delayed the decline in the spinal cord of SOD1-G93A mice in the mRNA levels of an essential transcription factor, PGC1α, a master regulator of mitochondrial biogenesis. The decrease in PGC1α in SOD1-G93A spinal cord was described before (Thau *et al*, 2012; Qi *et al*, 2015), but the mechanisms are unknown. A regulatory role on PGC1α was shown to occur downstream of Parkin-mediated regulation of PARIS (Siddiqui *et al*, 2015), since it was clearly demonstrated that PARIS degrades PGC1α in the nervous system and that Parkin is involved in regulating PARIS levels (Shin *et al*, 2011). We did not detect significant changes in PARIS content in the spinal cord of G93A or PKO/G93A mice, suggesting that in these disease models, PGC1α could be subjected to a different mode of regulation.

In conclusion, this work provides new evidence that Parkin is involved in MQC in the central nervous system under conditions of chronic mitochondrial damage. It offers an alternative perspective on the role of Parkin-mediated MQC in neurodegenerative diseases, whereby Parkin-mediated MQC could be neuroprotective in the short term, when damaged mitochondrial proteins, whole mitochondria, or mitochondrial components are removed and replaced. However, if mitochondrial damage continues unabated throughout the life of affected individuals, such as in the case of ALS, degradation of mitochondrial proteins and decreased mitochondrial biogenesis, leading to mislocalization and depletion of mitochondria, can hasten the neurodegeneration. While Parkin ablation temporarily delays this process, eventually compensatory mechanisms likely involving other mitochondrial ubiquitin ligases intervene to maintain MQC activity. In sum, we propose that MQC could be a double-edged sword in neurodegenerative diseases.

# Materials and Methods

## Reagents

All reagents are from Sigma-Aldrich (St. Louis, MO), unless stated otherwise.

## Animal studies

All animal procedures were approved by the Weill Cornell Medicine Animal Care and Use Committee and were performed in accordance with the Guidelines for the Care and Use of Laboratory Animals of the National Institute of Health. Mice were housed in a temperature and humidity-controlled facility on a 12-h light–dark cycle. Food and water were provided *ad libitum*. We used the following strains from The Jackson Laboratory (Bar Harbor, ME): B6SJL-Tg (SOD1*G93A)1Gur/J, B6.Cg-Tg(SOD1*G93A)1Gur/J (Gurney *et al*, 1994), and B6.129S4-*Park2*$^{tm1Shn}$/J (Goldberg *et al*, 2003). Mt-Keima (mitochondria-targeted Keima) mice were kindly provided by Dr. Toren Finkel (Sun *et al*, 2015). For the maintenance of the SOD1-G93A line in the B6SJ/L background, positive males (B6SJL-Tg (SOD1*G93A)1Gur/J) were bred with wild-type B6SJLF1/J females. Parkin KO (PKO) mice with the SOD1-G93A mutation (PKO/G93A) in the C57BL/6J congenic background were maintained by crossing heterozygous PKO females (B6.129S4-*Park2*$^{tm1Shn}$/J) with heterozygous PKO males carrying the SOD1-G93A mutation (B6.Cg-Tg (SOD1*G93A)1Gur/J). Mice were born at Mendelian ratios.

Mt-Keima animals expressing the SOD1-G93A mutation (mt-Keima/G93A) were obtained by crossing mt-Keima and B6.Cg-Tg (SOD1*G93A)1Gur/J males with mt-Keima females. Mt-Keima/G93A mice were born at the expected Mendelian ratios and their lifespan and disease course matched those of the C57BL/6J SOD1-G93A mice.

Primer sequences and protocols for genotyping of the offspring from tail DNA are available through The Jackson Laboratory for PKO and SOD1-G93A mice. For genotyping of the mt-Keima/G93A colony, we used the primers and amplification conditions reported before (Sun *et al*, 2015).

## Tissue collection and processing

A total of eight mice per group (four males and four females per group) were used unless otherwise indicated, with all mice paired according to date and gender. For biochemical purposes, mice were sacrificed by cervical dislocation at the indicated ages. Spinal cord, liver, and cerebellum were harvested immediately and placed in storage solution (10 mM Tris pH 7.4, 320 mM sucrose, and 20% (v:v) DMSO) and snap-frozen in liquid nitrogen. Samples were kept at −70°C, until use. A separate cohort of animals was dedicated to histological studies. Animals were terminally anesthetized with pentobarbital and then intracardially perfused with cold PBS, followed by 4% (w/v) paraformaldehyde (PFA) in PBS, using a peristaltic pump (flow speed 9 ml/min; 3 min for each solution). Afterward, tissue was collected and postfixed in 4% PFA overnight at 4°C. After several washes in PBS, spinal cords were dissected and placed in 30% sucrose in 0.1 M phosphate buffer. One week later, the lumbar spinal cords were removed from the solution and frozen at −20°C, until use. Soleus muscles were also dissected and postfixed in 4% PFA overnight. After washes in PBS, muscles were stored in 0.05% NaN$_3$ in PBS at 4°C until used.

For *in vivo* quantification of mitophagy, we used mt-Keima/G93A mice. Mt-Keima-labeled mitochondria in an acidic environment, such as autophagic vesicles or lysosomes, can be detected after excitation with the 543-nm laser, while healthy mitochondria, maintaining a higher pH, are detected after excitation with the

458-nm laser. Sequential images using both excitation wavelengths are recorded in an emission wavelength range of 600–650 nm, and the mitophagy rate was expressed as the percentage of mitochondrial area with high ratio (543/458 nm) signal normalized by total mitochondrial area (Katayama *et al*, 2011). For these studies, animals were sacrificed, spinal cords were freshly dissected, and sections were prepared for live imaging in a paired manner (i.e., one pair of mt-Keima/Non Tg and mt-Keima/SOD1-G93A). 200-μm-thick sections were obtained from the lumbar spinal cords in a McIlwain Tissue Chopper (Ted Pella, Inc.; Redding, CA). Sections were washed in PBS, mounted on microscope-grade cover glasses, and imaged immediately on a Leica TCS SP5 Confocal (Leica Microsystems Inc.; Buffalo Grove, IL). Images were analyzed with Metamorph software (Molecular Devices; Sunnyvale, CA).

### Survival analysis, clasping, and body weight

Body weight was measured weekly, starting at 45 days of age. Since PKO mice at this age have lower body weight (Palacino *et al*, 2004), independent of the SOD1-G93A transgene, for each animal, longitudinal body weight data were expressed as percentage of change relative to the weight at 45 days. The inability of a mouse to right itself when placed on its side (loss of righting reflex) was used as the defining criterion. Clasping or extension reflex was evaluated and scored as described previously (Choi *et al*, 2008; Weydt *et al*, 2003; Barneoud *et al*, 1997). Briefly, mice were suspended by the tail and the degree of motor deficit was scored from 0 to 5: No signs of clasping was scored 0; impaired extension on one side was scored 1; impaired extension bilaterally was scored 2; the absence of extension on one side was scored 3; the absence of extension bilaterally was scored 4; and total loss of response was scored 5.

A total of 18–20 mice per group were used. Equal numbers of males and females were used in each group. Special food, DietGel® Recovery (Clear $H_2O$; Westbrook, ME), was made available to all the mice under study at 135 days and kept until the endpoint.

### Mitochondria isolation

Frozen spinal cords were allowed to thaw on ice, washed in PBS to remove the excess of DMSO, and homogenized on ice in 1 ml of buffer H (10 mM Tris pH 7.4, 320 mM sucrose, 1 mM EDTA, 1 mM DTT, 1 mg/ml fatty acid-free bovine serum albumin (FAF-BSA), complete™ protease inhibitors cocktail (Roche Applied Science; Mannheim, Germany), and a mixture of phosphatase inhibitors (1 mM NaF, 1 mM $Na_3VO_4$, 1 mM pyrophosphate, and 2 mM imidazole). A total of 15 strokes in a glass-Teflon Potter Elvehjem-style homogenizer were used to disrupt the tissue. All further procedures were performed at 4°C. The resulting homogenate (*homogenate*) was centrifuged at $1,500 \times g$ for 5 min. The supernatant was collected and saved for later (*S1*) and the pellet was resuspended for additional homogenization, in buffer H containing FAF-BSA for 10 passes. After centrifugation at $1,500 \times g$ for 5 min, the supernatant was collected and combined with the previous one (*S1 + S1'*). This fraction was centrifuged at $15,000 \times g$ for 20 min. The pellet was washed in buffer H without FAF-BSA, followed by centrifugation in the same conditions ($15,000 \times g$ for 20 min). The supernatant was discarded, and the pellet of mitochondria was resuspended in buffer H without FAF-BSA for further experiments.

### Tissue lysis

Mitochondria or homogenates (tissue or cells) were combined in a ratio (1:1) with lysis buffer containing 20 mM HEPES pH 7.4, 100 mM NaCl, 100 mM NaF, 1 mM $Na_3VO_4$, 5 mM EDTA, 1% (v: v) Triton X-100, 0.5% (v:v) NP-40, complete™ protease inhibitor cocktail, and phosphatase inhibitors (1 mM NaF, 1 mM $Na_3VO_4$, 1 mM pyrophosphate, and 2 mM imidazole; Palomo *et al*, 2011) and were incubated on ice for 15 min. The soluble fraction (this applies to tissue and cell homogenates but not to mitochondria) was collected after centrifugation at 16,000 *g* for 10 min at 4°C. Then, the samples were mixed with Laemmli buffer containing SDS and heated at 70°C for 20 min (mitochondria) or boiled at 95°C for 5 min (cell and tissue homogenates). Protein concentration in the samples was measured with the DC™ Protein Assay (Bio-Rad; Hercules, CA).

### Western blots

All reagents, devices, and software used were from Bio-Rad (Hercules, CA) unless otherwise stated. Samples (30 μg of protein) were separated by gel electrophoresis in the presence of SDS on 15% acrylamide-bisacrylamide gels and on 8–12% and Any kD™ Mini-PROTEAN® TGX™ precast polyacrylamide gels. The proteins were electro-transferred to nitrocellulose or PVDF (Immun-Blot PVDF) membranes following standard procedures or using the Trans-Blot® Turbo™ transfer system. PVDF membranes used for ubiquitin detection were denatured by boiling in PBS for 10 min. Then, membranes were blocked with 5% (w:v) non-fat-dried milk prepared in PBS plus 0.1% Tween-20 (PBST). Subsequently, the filters were incubated overnight with the primary antibody diluted in PBST at 4°C and with constant agitation. After washes in PBST, the membranes were incubated with the corresponding peroxidase-conjugated secondary antibody for 1 h at room temperature. After additional washes in PBST, the antibody binding was visualized using the Clarity™ Western ECL blotting substrate and images were acquired and quantified with the ChemiDoc™ Touch imaging system and Image Lab software.

The following antibodies were used in Western blot (additional information can be found in Table EV1): Parkin (clone PRK8, 1:1,000, cat. # sc-32282, Santa Cruz Biotechnology; Dallas, TX), β-actin (clone AC-74, 1:1,000–1:5,000, cat. # A5316, Sigma), p62 (clone 2C11, 1:1,000, cat. # H00008878-M01, Novus Biologicals; Littleton, CO), optineurin (OPTN, 1:1,000, cat. # 10837-1-AP, Proteintech; Rosemont, IL), Complex V (ATP synthase beta, clone 3D5AB1, 1:1,000–1:5,000, cat. # A21351, Thermo Fisher Scientific; Waltham, MA), Tim23 (clone 32/Tim23, 1:1,000, cat. # 611222, BD Transduction Labs; San Jose, CA), COX1 (Complex IV, subunit 1, clone 1D6E1A8, 1:2,000, cat. #459600, Thermo Fisher Scientific), SOD1 (1:1,000, cat. # 574597, EMD Millipore; Billerica, MA), VCP (valosin-containing protein, clone 5, 1:1,000, cat. # MA3-004, Thermo Scientific; Rockford, IL), LC3 (microtubule-associated protein 1A/1B-light chain 3B, LC3B, 1:1,000, cat. # L7543, Sigma), March5 (membrane-associated ring-CH-type finger 5, 1:1,000, cat. # PA5-25584, Thermo Fisher Scientific), Mul1 (mitochondrial E3-ubiquitin protein ligase 1 1:1,000, cat. # HPA017681, Sigma), Miro1 (mitochondrial Rho GTPase 1, clone 4H4, 1:1,000, cat. # H00055288-M01, Novus Biologicals), mitofusin 2 (Mfn2, 1:1,000, cat. # M6319, Sigma), PARIS (parkin-interacting substrate

(ZNF746), clone N196/16, 1:1,000, cat. # 75-195, NeuroMabs and Antibodies Inc.; Davis, CA), ubiquitin Lys48-specific (clone Apu2, 1:1,000, cat. # 05-1307, EMD Millipore), ubiquitin Lys63-specific (clone Apu3, 1:1,000, cat. # 05-1308, EMD Millipore), and horse-radish-peroxidase (HRP) goat anti-rabbit, goat anti-mouse, and donkey anti-sheep secondary antibodies (1:10,000, Jackson ImmunoResearch; West Groove, PA).

### RNA extraction and reverse transcription

Total RNA was extracted from spinal cord samples using the RNAqueous®-4PCR kit (Ambion & Thermo Fisher Scientific), according to manufacturer's instructions. RNA concentration and purity were determined by absorbance (abs) at 260 nm and by the ratio abs260/abs280, respectively, using a Nanodrop spectropho-tometer (Thermo Fisher Scientific). 100 ng of RNA was reverse-tran-scribed using the ImProm-II™ Reverse Transcription kit (Promega; Madison, WI) to generate cDNA, according to manufacturer's instructions. Random hexanucleotide primers and 4 mM $MgCl_2$ (both provided with the kit) were used for the reaction. Samples of cDNA were kept at −20°C until further use.

### Real-time polymerase chain reaction (qPCR)

Gene expression, using cDNA as template, was measured with the QuantStudio 6 Flex Real-Time PCR System (Applied Biosystems Inc., Foster City, CA), using 96-well plates (Applied Biosystems) and Maxima SYBR Green/ROX qPCR master mix (Thermo Fisher Scientific). Standard curves were obtained to determine the effi-ciency of the primers (in a range 90–105%), together with melting curves and agarose-gel electrophoresis of the products to assess the specificity of the qPCR reaction. Comparative quantification method, based on the ΔΔCt algorithm of relative quantification (fold change) and with a calibrator sample and a housekeeping gene (β-actin), was used to express the results. The sequences and annealing temperatures of the primers used are (additional informa-tion can be found in Table EV2) as follows: mouse Parkin for 5′-AAGAAGACCACCAAGCCTTGTC-3′, mouse Parkin rev 5′-CAAA CCAGTGATCTCCCATGC-3′, 55°C (Wang et al, 2013); mouse PGC1α for 5′-TGAAAAAGCTTGACTGGCGTC-3′, mouse PGC1α rev 5′-CGCTAGCAAGTTTGCCTCAT-3′, 58°C (designed in the labora-tory); mouse β-actin for 5′-CTTTGCAGCTCCTTCGTTGC-3′ mouse β-actin rev 5′-CCTTCTGACCCATTCCCACC-3′, 55–58°C (designed in the laboratory).

### Immunohistochemistry

Fixed spinal cords were processed with the MultiCord® Technology (NeuroScience Associates, NSA; Knoxville, TN). 1-cm-thick lumbar segments of the spinal cords were sectioned in 40-μm-thick serial coronal sections. A total of 40 spinal cords were processed at the same time. Sections were kept in storage solution (from NSA) at −20°C until used. Spinal cord sections were immunostained as described before (Marques-Lopes et al, 2014) with minor modifications, and all the procedures were carried out with mild agitation. Briefly, slices were washed in 0.1 M phosphate buffer (PB) and blocked in 1% (w/v) BSA in 0.1 M phosphate buffer for 2 h at 4°C. Afterward, the tissues were incubated with the primary antibodies diluted in 0.1 M phosphate buffer supplemented with 0.5% (w:v) BSA and 0.25% (v:v) Triton X-100 (PB-BSA-T). Tissues were kept in primary antibody solution for 48 h at 4°C. After thorough washes, the secondary antibodies diluted in PB-BSA-T were added to the sections and incubated at room temperature for 2 h. Finally, sections were mounted in gelatin-coated slides using Fluoromount G (SouthernBiotech, Birmingham, AL) and allowed to dry for 24 h at room temperature. Samples were kept in the dark at 4°C until examined by confocal microscopy (Leica SP5). The following antibodies were used: GFAP (glial fibrillary acidic protein, cat. # Z0334, Dako & Agilent Technologies, Inc.; Santa Clara, CA), p62 (cat. # H00008878-M01, Novus Biologicals), ChAT (choline acetyltransferase, cat. # NBP1-30052, Novus Biologicals), Cy3 donkey anti-mouse, Cy2 donkey anti-rabbit, and Cy2 donkey anti-goat (Jackson ImmunoResearch). All images were analyzed in Metamorph. For GFAP, total fluorescence intensity, using the integrated morpho-metry analysis module and inclusive threshold, was quantified with a region of interest (ROI) of 250 μm diameter that was drawn in the ventral horn of each spinal cord. Similarly, p62 aggregates were quan-tified with the integrated morphometry analysis module with inclusive threshold, by drawing regions containing the gray matter for each spinal cord. ChAT-positive motor neurons were counted manually. A total of eight to ten mice (same number of males and females) were used for each genotype.

Neuromuscular junctions (NMJ) were studied in soleus muscles. Briefly, fixed muscles were washed in PBS and blocked overnight at 4°C in blocking buffer composed of 5% (v:v) normal goat serum, 2.5% (w:v) BSA, and 1% (v:v) Triton X-100 in PBS (PBS-NGS-BSA-T) with mild agitation. Subsequently, an antibody against neurofila-ment H (NF-H, to label the motor axon terminals; cat. # AB1989, Millipore) diluted in PBS-NGS-BSA-T was added to the tissues and kept for 48 h at 4°C. After washes in PBS, the tissues were incu-bated with the secondary antibody Cy2 goat anti-rabbit (Jackson ImmunoResearch) diluted in PBS-NGS-BSA-T and with tetramethyl-rhodamine-labeled α-bungarotoxin (α-BTX, Thermo Fisher Scien-tific) to label the acetylcholine receptors in the motor endplates, overnight at 4°C. After thorough washes, the muscles were mounted in microscopy-grade glasses using Fluoromount G and kept at 4°C in the dark until imaged by confocal microscopy. Images were analyzed in Metamorph and counted manually. A total of 325–483 NMJ were counted, from six different mice (three males and three females) per genotype.

### Mitochondrial enzyme activities

Citrate synthase (CS) and cytochrome c oxidase (COX) activities were measured in homogenates (in S1 fraction) from spinal cords as previously described (Birch-Machin & Turnbull, 2001), with minor modifications described in www.oxphos.org. For the measurements, 10 μg of protein was used.

### Filter-trap assay

The presence of large insoluble SOD1 aggregates was determined by filter-trap assays as described before (Wang et al, 2002). Briefly, 10 μg of protein samples (S1 and mitochondrial fractions) was treated with 0.5% (v:v) Triton X-100 for 15 min on ice prior to loading in a Bio-Dot® Microfiltration apparatus (Bio-Rad). Cellulose acetate membranes (0.2 μm pore diameter, Whatman Gmbh; Dassel,

## The paper explained

### Problem

Amyotrophic lateral sclerosis (ALS) is a devastating neuromuscular disorder that causes paralysis and death. The first discovered ALS gene was CuZn, superoxide dismutase (SOD1). Mutations in SOD1 cause motor neuron degeneration and ALS in humans and in transgenic mice. SOD1 mutations result in extensive mitochondrial damage in the spinal cord, but it is not known how affected cells cope with the accumulation of damaged mitochondria.

### Results

We have investigated mitochondrial quality control in the spinal cord of transgenic mice expressing mutant SOD1 (SOD1 G93A mice). We show that mitochondrial degradation is activated in mutant mice, leading to increased mitophagy (i.e., mitochondrial autophagy). We find that genetic ablation of Parkin, a mitophagy-initiating protein, ameliorates the disease in SOD1 G93A mice by delaying key pathological events in the spinal cord and the denervation of the neuromuscular junction and significantly extends survival.

### Impact

We propose that modulation of the mitochondrial quality control pathways in ALS spinal cord has a positive impact on the disease by delaying mitophagy and the associated degradation of mitochondrial proteins involved in mitochondrial dynamics.

Germany) were used for the assay, and samples were allowed to pass through the filter by applying vacuum. After several washes, membranes were blocked in 5% (w:v) non-fat-dried milk prepared in PBS plus 0.1% Tween-20 (PBST). Anti-SOD1 (1:1,000) antibody was used for detection.

### NSC34 cell culture, transfection, and mitoPQ treatments

NSC34 motor neuron-like cells stably expressing G93A mutant SOD1 or empty vector control (Magrane *et al*, 2009) were grown in high glucose (25 mM) Dulbecco's modified Eagle's medium (DMEM, Thermo Fisher Scientific; Waltham, MA) supplemented with 10% (v: v) fetal bovine serum (FBS, Atlanta Biologicals; Flowery Branch, GA), 4 mM glutamine (Thermo Fisher Scientific), and 500 μg/ml G418 (Thermo Fisher Scientific), in a humidified incubator at 37°C and 5% $CO_2$. Mycoplasma contamination was routinely excluded by PCR of the culture medium, using specific primer sets. Cells were subcultured every 3–4 days. NSC34 cells were transfected with either YFP-Parkin (Takara Bio USA Inc.; Mountain View, CA) or pEGFP-N1 (Addgene plasmid 23955; Cambridge, MA) and were then exposed to mito Paraquat (mitoPQ; Robb *et al*, 2015) for 24 h. Cell viability was ascertained by comparing the number of residual transfected cells (i.e., YFP-Parkin or GFP positive), with and without mtPQ treatment. A total of 38–40 imaging fields for each condition were evaluated. The results were expressed in percentage of vehicle (DMSO)-treated cells.

### Statistics

Data are presented as mean ± SEM. Kaplan–Meier survival analysis and two-way ANOVA with repeated measures (for body weight analysis) with Holm–Sidak's correction for multiple comparisons were run in Sigma Plot (Systat Software; San Jose, CA). The rest of the data were analyzed in GraphPad (GraphPad Software Inc.; La Jolla, CA).

All datasets were tested for normal distribution with the D'Agostino–Pearson's, Shapiro–Wilk's and Kolmogorov–Smirnov's tests. As a general rule, when two samples were analyzed in parallel, we considered them experimental pairs and, if more than two samples were analyzed, we considered them groups of randomized block experiments. Data used as reference (i.e., Non Tg) were set at 100%. The data groups in the experiments were normally distributed, unless specified. *P*-values < 0.05 were considered significant.

**Expanded View** for this article is available online.

## Acknowledgements

We thank Dr. Toren Finkel (NIH NHLBI) for providing the mt-Keima mouse line, the Weill Cornell Medicine Veterinary Services for their help with the mouse colonies, and the BMRI Neuroanatomy Electron Microscopy Core for their invaluable help with the histology experiments. We also thank Dr. Alfredo Gimenez-Cassina for discussing the results and providing helpful advice on the manuscript. We also thank the Confocal Microscopy Core from CBMSO, and Dr. Javier Díaz-Nido and Mauro Agro (Madrid, Spain) for their support with the fluorescence quantification. Funding for this project was from NIH/NINDS grant R01NS062055 (to G.M.).

## Author contributions

GMP designed, performed, and interpreted experiments and wrote the manuscript; HK designed and performed experiments; VG designed and performed experiments; CK designed and performed experiments; MK performed experiments; AJA performed experiments; DZ performed experiments; TAM designed experiments; GM designed and interpreted experiments and wrote the manuscript.

## Conflict of interest

The authors declare that they have no conflict of interest.

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
