## [Review Process File · EMBO Molecular Medicine]

Parkin is a disease modifier in the mutant SOD1 mouse model of ALS

Gloria M. Palomo, Veronica Granatiero, Hibiki Kawamata, Csaba Konrad, Michelle Kim, Andrea J. Arreguin, Dazhi Zhao, Teresa A. Milner, Giovanni Manfredi

Review timeline:

Submission date:	22 January 2018
Editorial Decision:	08 March 2018
Revision received:	19 June 2018
Editorial Decision:	19 July 2018
Revision received:	26 July 2018
Accepted:	27 July 2018

Editor: Céline Carret

Transaction Report:

1st Editorial Decision

08 March 2018

Thank you for the submission of your manuscript to EMBO Molecular Medicine. We have now heard back from the three referees whom we asked to evaluate your manuscript.

You will see from the comments of the referees pasted below, that they find the study to be of interest and deserving publication. However, they also suggest additional experiments, details and clarifications to increase the conclusiveness and strengthen the data.

We therefore would welcome the submission of a revised version within three months for further consideration and would like to encourage you to address all the criticisms raised as suggested to improve conclusiveness and clarity. Please note that EMBO Molecular Medicine strongly supports a single round of revision and that, as acceptance or rejection of the manuscript will depend on another round of review, your responses should be as complete as possible.

I look forward to receiving your revised manuscript.

***** Reviewer's comments *****

Referee #1 (Remarks for Author):

While acute mitochondrial damage activates the mitochondria quality control (MQC), which clears damaged mitochondria, a chronic damage such that occurring in ALS could result in prolonged activation of the MQC and depletion of essential mitochondrial components due to excessive turnover. Authors of this manuscript have demonstrated using *in vivo* genetic crossing that increased mitochondria quality control activity in the spinal cord of the mutant SOD1 mouse model of ALS results in increased Parkin turnover to deplete mitochondria, ultimately causing a worsening of the neurodegeneration in ALS.

This study clearly points to a specific mitochondria-related mechanism as potential modifier of disease, which can eventually be targeted in follow-up pre-clinical therapeutic studies. There are several strengths here, including a thorough introduction and discussion sections, rigor in the experimental approach and execution of the experiments.

Several weaknesses and points of concern have also been noted, which require both incorporation of new experiments and clarifications.

General Comments:

- The crucial findings of the manuscript such as the measure of the spinal cord levels of Parkin across different disease stages and between the mouse genotypes or the mitophagy flux determination (Figs 1, 2, 3) should be extended to CNS areas non-affected by neurodegeneration and/or non-CNS tissues such as liver or muscle groups. This is important to clearly establish whether mitophagy and Parkin dysregulation is a true disease modifier occurrence;
- Further authentication of the role of Parkin as disease modifier could also come from the analysis of the disease phenotype in mutant SOD1 mice that overexpress the Parkin gene. In this case, it is expected a worsening of the ALS phenotype.

Points of concern

- Fig.2 clearly shows that mitophagy increases in the control mice as well, although at a slower pace compared to the mutant SOD1 mice. This increase in control mice is not or should not be related to an aging phenomenon as mitophagy has been assessed in relatively young adult mice. Perhaps, analysis of mitophagy using age-matched non-transgenic control mice would clarify whether this increase in mitophagy is physiological or it is caused by the overexpression of the human transgene, either wild-type or SOD1-G93A mutant.
- Fig.3 is most concerning. Expression of Parkin appears to be greatly downregulated or even depleted already at 60 days, the earliest time point analyzed, and does not further decrease over time in the mutant SOD1 mouse. Rather the decrease in Parkin is seen in a cross-section analysis with the WT-SOD1 mouse. Analysis Parkin levels should be extended to earlier presymptomatic time points (30 - 60 days interval) to establish whether the mutant SOD1 mouse displays constitutively depleted Parkin levels compared to normal control non-transgenic or WT-SOD1 mouse or whether Parkin is indeed depleted earlier. The implication of these data is important to establish whether Parkin is indeed a modifier or rather a triggering event in the degeneration of motor neurons.
- Parkin shows a different western blot profile in the 2 mouse backgrounds (B6 vs B6SJL). See Fig.3A compared to Fig. 4A. This might be because of difference in exposure time or difference in the antibody concentration used (non specified). Similarly, the Parkin/Beta-actin ratio represented in these figures is discordant, off by ~ 10 fold. This is concerning.
- It would be informative to repeat the time-course of Parkin depletion in the B6 bigenic mouse, over the three stages of disease as it was previously reported for the B6SJL mouse.
- Fig5D-F: Harmonize the scale of the graphs with the one used for Fig.1B,D.
- Weight loss is a late indicator of disease onset. Use an earlier indicators to gauge onset of symptoms, such as gait defect, and determine whether Parkin levels manipulation affects disease duration.
- Fig.7 and 8. The effect would be better appreciated if normalized to controls and expressed as % variation

Referee #2 (Remarks for Author):

In this article, authors investigated the role of Parkin mediated mitophagy in SOD1 G93A mutation induced progression of ALS. In a nutshell, this study comes to the conclusion that Parkin mediated mitophagy in the late stages of the disease is detrimental. Accordingly, knocking out Parkin proves to be beneficial for slowing down the disease progression. Overall the experimental data support many of the conclusions drawn by the author. However the authors fail to explain how Parkin KO can be protective and whether compensatory effects due to germline disruption in Parkin might account for the protective effect that they reported (rather than Parkin KO itself). Having said that, I think this is a very interesting work, which highlights in an *in vivo* model of neurodegeneration the potential dark side effect of exacerbated mitophagy. In details:

1. The study lacks the explanation how Parkin down regulation can increase mitophagy and Parkin knock out can enhance neuronal survival, especially when there is no compensation from the other E3-Ub ligases. Fig.1 shows that P62 and OPTN go up on 120d in mitochondrial fraction and Fig. 3 shows that Parkin level goes down both globally and in mitochondrial fraction. So it demonstrates a reverse correlation between the Parkin level and autophagy adaptor proteins. However, this correlation was perturbed in the Parkin KO G93A mice, where there was no Parkin, but OPTN and P62 levels were down. One might expect a steady correlation between the mitophagy marker and Parkin level, or it might raise the question how reliable the marker is. If the authors support the idea that Parkin recruits OPTN and p62 to mitochondria (as they have mentioned in the result section), they did not explain why in 120d old mice there was lower mitochondrial Parkin but high OPTN/p62. One explanation is reduced mitophagy, but it is not reflected by COX1 and TIM23 data (Fig. 1E). Interestingly, OPTN was unchanged in non TG and G93A in Fig. 5C and D.

2. The mitochondrial inner membrane protein markers selected to support the hypothesis that there was increased mitophagy (COX1 and TIM23) also raises the doubt because mitokema data (Fig.2) shows high level of mitophagy in 60 and 90d old mice, but was reflected only at 120d when the protein levels were considered. Of course these are two different approaches, but if they don't complement each other it is better to consider only one of them, or use some other mitochondrial marker, for example Cyclophilin D, which is a matrix resident protein. The use of TIM23 is not ideal since the mitochondrial import apparatus has been reported to be affected in ALS. In addition, the use of COX1 raises further doubts because though in Non-TG mice, mitophagy is apparently increasing age dependently (Fig.2C), this was not reflected in the COX1 levels (where it was age dependently increased, Fig.1E). One explanation is simultaneous increased mitochondrial clearance and generation. But that should be supplemented, at least, with PGC1 alpha level.

3. In the discussion section authors mentioned that the Parkin KO mice does not develop severe neurodegeneration due to compensatory mechanisms. As already pointed out at the beginning, they need to address whether the delay in SOD1G93A mediated phenotypes, is not because of those compensatory pathways and how it is directly modulated by Parkin. In that respect the use of Parkin conditional KO that was generated by the group of Ted Dawson could be of great help.

4. It is notable that the G93A mice starts showing muscle weakness and weight loss from 130 days, when mitophagy levels were found to be unchanged (120d, Fig. 2C). Authors need to address this if they want to conclude that increased mitophagy is involved in the disease progression.

5. Mitophagy was not measured and compared in the PKO and PKO/SOD1 G93A groups, at least by measuring the standard mitochondrial protein markers.

6. The discussion part does not always support the author's findings. "We also detect accumulation of mitochondrially localized MQC proteins, p62 and OPTN, suggesting that ultimately the ability to eliminate damaged mitochondria through mitophagy in SOD1-G93A spinal cord is compromised." - this statement is not supported by the mitokema data as there was no difference with the non TG group on 120d. Then they mentioned, "Together, these results indicate that Parkin depletion is due to increased protein turnover, as a result of hyperactive MQC." This is confusing.

Minor concerns

1. Maybe it is a misunderstanding, but they mentioned in the first paragraph of result section that the SOD1-G93A mice have an average life span of 129 days, but in the survival graph or in the body weight graph no effect is seen before 140 days.
2. In the Fig. 1A, the nonTG OPTN bands do not reflect the quantification.
3. Purity of the mitochondrial fraction is a matter of concern, especially in the LC3 blot where a high level of LC3-1 was also present in the mitochondrial fraction.
4. Somehow the bar graphs (+/- SEM) do not always reflect the statistical significance.
5. The endogenous SOD1 shows high level aggregates in Fig. 4E, is there any difference between PKO and non TG? The explanation how Parkin knock down enhance the clearance of the aggregates is not clear. If the authors hypothesize that SOD1 aggregates in mitochondrial membrane act as the seed for clearance they need to show that.
6. Fig. 7C, D. Statistical analysis between non TG with the other groups are required, if the authors claim that Parkin KO has a role in determining the PGC1 alpha level.

Referee #3 (Remarks for Author):

The manuscript by Palomo et al., uses a well-characterized model of Amyotrophic Lateral Sclerosis (the mutant SOD1-G93A mouse model) to study the mechanisms underlying activation of the mitochondria quality control (MQC), leading to mitophagy and, eventually, neurodegeneration. The study shows that, in the spinal cord of the SOD1-G93A mice two MQC proteins, p62 and optineurin, are activated leading to mitophagy. Looking at the mechanisms of mitochondria dysfunction, the authors show that Parkin and other mitochondria proteins, namely Drp1, Miro1 and Mfn2 (all ubiquitinated by parkin) and PGC1alpha are also involved. Contrary to the expectations, crossing of the Parkin KO mice with the SOD1-G93A mice, protects against neurodegeneration and slows down disease progression in the ALS mice.

This is a well written manuscript from one of the best mitochondria groups. The lab has contributed significantly not only to our understanding of mitochondria biology, but also to the identification of mitochondria pathology and dysfunction in ALS. In fact, while many groups have provided evidence that mitochondria play a crucial role in ALS pathogenesis (and in SOD1-ALS in particular), this group has been at the forefront, leading the ALS field with their significant expertise in mitochondria.

As expected from this group, the biochemical analysis presented follows a rigorous experimental approach and it is based on a solid rationale. The novelty of this manuscript is in (a) the identification of parkin as-perhaps- a significant player in ALS and (b) the use of the mt-Keima/SOD1-G93A to study mitophagy dynamics *in vivo*.

However, the results are largely descriptive and there are several concerns that greatly diminish the enthusiasm for the work in its current format.

1. As expected from biochemical and immunological analysis from tissues, most of the data presented that rely on western blot or immunohistochemical analysis and quantification of mitochondria proteins, are not as solid as one would like to see to make definite conclusions. Few examples: (1) Figure 1A & B- The effects on OPTN levels over disease progression are small and there is a high variation of effects over time in the SOD1-G93A mice. While at 120 days, OPTN levels seem to increase (albeit a very modest-but significant increase), at 60 and 90 days they seem to decrease (is the difference between non tg and SOD1-G93A mice significant at 90 days?). One would expect either a gradual increase over time (that follows disease progression) or no change followed by a spike at 120 (as for p62-Figure 1 C-D). The authors do not comment on the lack of a gradual increase over time nor on the small (and inconsistent) effects of disease on OPTN expression. (2) FIG 1 E-F There is a discrepancy between the WB shown in Figure 1E and the

quantification shown in Fig 1G. Why do COX1 levels increase in WT-SOD1 at 120 days and, overall, increase over time as compared to control? In Figure 1F, the quantification (COX-1 is normalized to actin that does not seem to change so much in the WB) does not show higher levels of COX1 in 120 days old WT overexpressors SOD1 mice. (3) FIGURE 5 C, D,, E, F. It is puzzling that, contrary to what shown in Figure 1, the graph in Figure 5D, does not show significantly higher levels of OPTN in G93A mice compared to non-tg mice. Perhaps, as said above, with such small and inconsistent effects, are hard to capture putting in to question the relevance of the findings.

2. It is not clear why there is a mis-match in the time course of the mtKeima/SOD1-G93A mice shown in Figure 2 (2C0 and all the data related to MQC proteins in Figure 1.

3. In Figure 4D, since the authors are using a dot-blot analysis, they should include the WT-hSOD1 transgenic mice as control.

4. The data shown in Figure 6 are interesting. As it relates to Figure 6C, at what time do the authors start seeing NMJ defects and denervation? Does denervation precede or is it concomitant with changes in MQC proteins levels and parkin?

Overall, this is an interesting, but descriptive work.

1st Revision - authors' response

19 June 2018

Point by point response to reviewers' comments

First, we want to thank the reviewers for their thorough analysis of the work and their insightful comments. They provided important suggestions for improving the manuscript. In the revision, we have addressed their comments and concerns as follows. Please note that modified text is colored in blue throughout the revised manuscript.

Referee #1 (Remark for Authors)

While acute mitochondrial damage activates the mitochondria quality control (MQC), which clears damaged mitochondria, a chronic damage such that occurring in ALS could result in prolonged activation of the MQC and depletion of essential mitochondrial components due to excessive turnover. Authors of this manuscript have demonstrated using in vivo genetic crossing that increased mitochondria quality control activity in the spinal cord of the mutant SOD1 mouse model of ALS results in increased Parkin turnover to deplete mitochondria, ultimately causing a worsening of the neurodegeneration in ALS.

This study clearly points to a specific mitochondria-related mechanism as potential modifier of disease, which can eventually be targeted in follow-up pre-clinical therapeutic studies. There are several strengths here, including a thorough introduction and discussion sections, rigor in the experimental approach and execution of the experiments.

Several weaknesses and points of concern have also been noted, which require both incorporation of new experiments and clarifications.

Question 1: The crucial findings of the manuscript such as the measure of the spinal cord levels of Parkin across different disease stages and between the mouse genotypes or the mitophagy flux determination (Figs 1, 2, 3) should be extended to CNS areas non-affected by neurodegeneration and/or non-CNS tissues such as liver or muscle groups. This is important to clearly establish whether mitophagy and Parkin dysregulation is a true disease modifier occurrence.

Answer: To address this issue, Parkin protein levels were measured in cerebellum, a CNS area that is largely unaffected in ALS, and liver, a peripheral non-affected tissue, obtained from Non Tg, G93A, PKO, and PKO/G93A (in the C57BL6 background, n= 8 mice per genotype) at 130 days. Results are presented in Expanded View 1A-B and 1D-E. As expected, no Parkin expression was detected in either PKO or PKO/G93A mice. Parkin levels in liver were unchanged compared to Non Tg, but Parkin in G93A cerebellum was 20% lower than in the Non Tg controls. By comparison, at the same age, Parkin levels in spinal cord from G93A mice drop to 53% compared to Non Tg

(Figure 4A and 4B). This suggests that while Parkin is abnormal in cerebellum, the decline is not as pronounced as it is in the affected spinal cord, suggesting that Parkin decline relative to Non Tg proceeds in parallel with pathology. The panels illustrating these new data are also shown below for ease of review (please note, Cb denotes cerebellum). We have included these data also in the results

section.

We also looked at p62 in cerebellum and liver. There was no difference in p62 levels in liver, while in cerebellum p62 was increased in both PKO and PKO/G93A relative to Non Tg, but not in G93A mice (Expanded View 1C and 1F). This result confirms that p62 increases in a G93A mutant SOD1 dependent manner only in spinal cord. We have included these data in the results section, where we present the double transgenic mice.

Question 2: Further authentication of the role of Parkin as disease modifier could also come from the analysis of the disease phenotype in mutant SOD1 mice that overexpress the Parkin gene. In this case, it is expected a worsening of the ALS phenotype.

Answer: Mice expressing wt Parkin in spinal cord neurons are not currently available to us. There is a mouse line that Jackson Laboratories used to distribute, which expresses Parkin under the prion promoter. However, whether the transgene is being expressed in the spinal cord has not been tested in this line. The line is cryopreserved, but we will try to acquire it for future studies, since it would take a long time to revive the line, breed, and cross with G93A mice for this study. Nevertheless, we

Palomo GM et al. Expanded View 5

appreciate that this is an important point to address. Therefore, we adopted a cell culture approach. We used NSC34 motor neuron-like cells stably expressing G93A mutant SOD1 or vector control (mock). These cells do not express endogenous Parkin. Cells were transfected with either YFP-Parkin or pEGFP-N1 as a control. Cells were then exposed to a mitochondria-specific oxidative stress with mtPQ (a form of paraquat selectively targeted to mitochondria, Robb et al. Free Radic Biol Med, 2015) and we assessed YFP-Parkin distribution and the viability of the transfected cells. As expected, YFP-Parkin clustered with mitochondria upon mtPQ treatment. Viability data were normalized by cells not treated with mtPQ or treated with vehicle as a control. We found that mtPQ causes death only in mock cells expressing YFP-Parkin and not in cells expressing GFP. Toxicity associated with mtPQ and YFP-Parkin was more pronounced in SOD1-G93A expressing cells (Expanded View 5 A-C). These data indicate that Parkin expression enhances cell death, when mitochondria are damaged. The panels illustrating these new data are shown here for ease of review. We have included these data in the results section.

Kymographs of mitochondrial labeled with mitoDsRed and quantification of motile mitochondria in distal and proximal segments of axons of motor neuron obtained from embryonic spinal cord of Non Tg mice. n=10-12 axons.

We have also tested the effects of YFP-Parkin expression on mitochondrial motility of primary embryonic spinal motor neurons. We performed kymographs of mitochondria labeled with mitoDsRed to estimate the proportions of moving and stationary mitochondria. The expression of Parkin decreased the proportion of moving mitochondria in both Non Tg and SOD1-G93A neurons. These results are in agreement with the hypothesis that Parkin decreases mitochondrial motility in neurites, possibly because it facilitates the turnover of mitochondrial dynamics proteins, such as Miro1. We have only performed this experiment in a very limited number of motor neurons (10-12) and therefore at this time we prefer to include the results only in this response for reviewer's inspection. Future work will further investigate mitochondrial motility and distribution in motor neurons of Parkin overexpressing mice with and without mutant SOD1.

Question 3:

Fig.2 clearly shows that mitophagy increases in the control mice as well, although at a slower pace compared to the mutant SOD1 mice. This increase in control mice is not or should not be related to an aging phenomenon as mitophagy has been assessed in relatively young adult mice. Perhaps, analysis of mitophagy using age-matched non-transgenic control mice would clarify whether this increase in mitophagy is physiological or it is caused by the overexpression of the human transgene, either wild-type or SOD1-G93A mutant.

Answer: We apologize for the lack of clarity. Wild type SOD1 Tg mice were not used in this experiment. The SOD1-G93A mice are compared to Non Tg littermates, which do not overexpress

SOD1. Statistical analysis showing the variations over time has been incorporated into Figure 2C. This new analysis shows that mitophagy increases at a similar pace over time in Non Tg and G93A mice. The fact that both genotypes present a gradual increase in mitophagy over time could be related to neurodevelopment, when adjustments in mitochondria content and localization may be required for motor neuronal maturation and growth. What we think is remarkable here is that G93A have higher mitophagy than Non Tg until they reach the disease stage. A plausible explanation for the inability of motor neurons to boost mitophagy at symptomatic stage could be the exhaustion of essential components of the machinery, such as Parkin. The revised figure 2C with the added statistics is shown here.

Question 4 :

Figure 3. Expression of Parkin appears to be greatly downregulated or even depleted already at 60 days, the earliest time point analyzed, and does not further decrease over time in the mutant SOD1 mouse. Rather the decrease in Parkin is seen in a cross-section analysis with the WT-SOD1 mouse. Analysis Parkin levels should be extended to earlier presymptomatic time points (30 - 60 days interval) to establish whether the mutant SOD1 mouse displays constitutively depleted Parkin levels compared to normal control non-transgenic or WT-SOD1 mouse or whether Parkin is indeed depleted earlier. The implication of these data is important to establish whether Parkin is indeed a modifier or rather a triggering event in the degeneration of motor neurons.

Answer: To address the reviewers question we have added to Figure 3A and 3B western blots and quantifications of Parkin in spinal cord homogenates from SOD1-G93A and Non Tg at 30 days of age. To quantify Parkin levels, we use fold change relative to Non Tg samples collected at 30 days as normalizer values. This allows us to directly compare the results obtained from 8 mice for each genotype and better detect average changes over time. The results for 30 days are now included and the samples corresponding to Non Tg mice at 30 days are used for normalization. The progressive increase in Parkin levels in Non Tg mice does not occur in SOD1-G93A mice. A trend for lower Parkin in G93A starts to be appreciated at 60 days, but it only reaches statistical significance at 90 days and 120 days (disease onset and end stage, respectively). However, if we analyze the results using a different approach, which involves a separate analysis of pairs for each time point, with the Non Tg values used as internal reference for each time point (i.e., 100%), we clearly see that Parkin levels differ at 90 days in G93A, and not at 60 days, and is sustained at 120 days. Both analyses indicate that Parkin protein is not affected in the pre-symptomatic stages of the disease and the decline starts in the symptomatic phase. However, it has been widely reported that pathological changes in motor neurons can be observed in pre-symptomatic phases of the disease. This would suggest that Parkin acts more as a disease modifier than a triggering factor. We added another set of

western blots (out of 8 sets) and the quantification of the data in % for each time point for reviewer's inspection. In the manuscript figure (Fig. 3), we have added the 30 day time point, but we express all the data as % of 30 days Non Tg for consistency with the other figures.

Question 5:

Parkin shows a different western blot profile in the 2 mouse backgrounds (B6 vs B6SJL). See Fig.3A compared to Fig. 4A. This might be because of difference in exposure time or difference in the antibody concentration used (non-specified). Similarly, the *Parkin*: β -actin ratio represented in these figures is discordant, off by ~ 10 fold. This is concerning.

Answer: *Parkin* protein was always detected with the antibody PRK8, from Santa Cruz Biotechnology, at a concentration of 1:1000, diluted in 5% milk and with O/N incubation at 4°C. The differences seen between the western blots presented on Figures 3A and 4A can be attributed to different acrylamide percentage used for running the gels. While in Figure 3A, due to the need for detecting several other proteins with high molecular weights in the same blot, 7.5% acrylamide gels were used, for Figure 4A and 3D we used commercial AnyKD gels (from BioRad), which are optimal for the resolution of small to middle size polypeptides. We do not think this different band pattern could be related to alternative splicing, because the same starting material was used to first prepare homogenates and then mitochondria from the homogenates (Figures 3A and 3D, respectively) and the banding pattern is different, due to gel electrophoresis. The two gels are shown here for comparison.

To follow the reviewer's advice, we now express all *Parkin* protein levels (as well as most other proteins measured in western blots) relative to the earliest time point in Non Tg controls. This is probably the best way to quantify, since sample sets of 8 were run (G93A and Non Tg B6SJL at each time point) and immunodetection was carried out in the same membrane, so we could compare differences between the genotypes and during the time course of the disease. However, for the PKO/G93A (C57BL6) colony, replicates of the four genotypes for each time point were run in pairs (8 samples), so we could quantify inter-genotype differences, but in this case not throughout the disease time course. If we analyze each time point separately in the B6SJL mice, using the Non Tg control as normalizer, *Parkin* levels in the G93A mice at 90 days correspond to 31% of Non Tg control. In the C57BL6 at 130 days, *Parkin* was 53% of Non Tg control (Figure 4A and 4B). Thus, values are within a similar range, although there is a difference that can be attributed to timing differences, as in the double transgenic PKO/G93A the timing of the disease stages could be slightly modified. In the text, we now indicate that western protein levels are shown as percentage of the control, and the antibody concentrations used for immune-detection have been added in the Material and Methods section.

Question 6:

*It would be informative to repeat the time-course of *Parkin* depletion in the B6 bigenic mouse, over the three stages of disease as it was previously reported for the B6SJL mouse.*

Answer: Here we show biochemical data obtained from mice at 130 days (onset). The reason why we opted to focus on this point is because the genetics of the mouse with the three alleles (biallelic PKO plus G93A) requires a lot of breeding. We ensured that we studied 8 mice per genotype at each time point equally divided between males and females. This effort took approximately 18 months of breeding, not including the work done on the B6SJL. We understand the reviewer's point, but we had to deal with these constraints of the mouse genetics. Therefore, we feel that as much as we

would like to look at all the time points, it would be a very lengthy effort right now. We have looked at a pair of SOD1-G93A and Non Tg mice in the C57BL6 background at 30 days and found no discernible difference at this age in spinal cord Parkin level. We show this western blot here. Clearly, this experiment is still too limited to make a firm conclusion, but it agrees with the findings in the pre-symptomatic B6SJL SOD1-G93A mice.

Question 7:

Fig5D-F: Harmonize the scale of the graphs with the one used for Fig.1B, D.

Answer: Following the reviewer's suggestion, we now express all protein levels as percentage of Non Tg controls. Please, note that while at 130 days C57BL6 G93A mice are at disease onset,

B6SJL G93A mice are near end stage. Initially, we decided to show the results of C57BL6 at 130 days because the end stage in PKO/G93A mice is nearly 25 days later than in G93A, and because both genotypes carrying the G93A mutation show increased levels of the mitophagy adaptors at end stage. We believe that early time points showing differences between both genotypes would be more informative in terms of disease pathogenesis. We have now included in Expanded View 3 the western-blot images and quantifications for p62, VCP and OPTN at disease end stage in the C57BL6 G93A mice (also shown here). The results are consistent and reproduce the results observed in B6SJL mice, reported in Fig. 1 A-B, which showed accumulation of p62 mitophagy adaptor in mitochondria.

Question 8:

Weight loss is a late indicator of disease onset. Use an earlier indicator to gauge onset of symptoms, such as gait defect, and determine whether Parkin levels manipulation affects disease duration.

Answer: We have added clasping measurement as a much earlier indicator of disease onset than

body weight loss. Clasping assessment shows no difference between G93A and PKO/G93A. This is consistent with the conclusion that PKO does not affect disease onset of G93A mice, as much as it affects disease progression and survival. Clasping score has been assigned as follows: No signs of clasping= 0; imbalance extension in one side= 1; imbalance extension bilaterally=2; absence of extension in one side= 3; absence of extension bilaterally= 4; paralysis=5. The graph of clasping is now presented in Figure 6C and also shown here.

Question 9:

Fig.7 and 8. The effect would be better appreciated if normalized to controls and expressed as % variation.

Answer: As suggested by the reviewer, we now express these values as a % of Non Tg (see also response to point 5).

Referee #2 (Remarks for Author):

In this article, authors investigated the role of Parkin mediated mitophagy in SOD1 G93A mutation induced progression of ALS. In a nutshell, this study comes to the conclusion that Parkin mediated mitophagy in the late stages of the disease is detrimental. Accordingly, knocking out Parkin proves to be beneficial for slowing down the disease progression. Overall the experimental data support many of the conclusions draw by the author. However, the authors fail to explain how Parkin KO can be protective and whether compensatory effects due to germline disruption in Parkin might account for the protective effect that they reported (rather than Parkin KO itself). Having said that, I think this is a very interesting work, which highlights in an in vivo model of neurodegeneration the potential dark side effect of exacerbated mitophagy. In details:

Question 1.

The study lacks the explanation how Parkin down regulation can increase mitophagy and Parkin knock out can enhance neuronal survival, especially when there is no compensation from the other E3-Ub ligases. Fig.1 shows that P62 and OPTN go up on 120d in mitochondrial fraction and Fig. 3 shows that Parkin level goes down both globally and in mitochondrial fraction. So it demonstrates a reverse correlation between the Parkin level and autophagy adaptor proteins. However, this correlation was perturbed in the Parkin KO G93A mice, where there was no Parkin, but OPTN and P62 levels were down. One might expect a steady correlation between the mitophagy marker and Parkin level, or it might raise the question how reliable the marker is. If the authors support the idea that Parkin recruits OPTN and p62 to mitochondria (as they have mentioned in the result section), they did not explain why in 120d old mice there was lower mitochondrial Parkin but high OPTN/p62. One explanation is reduced mitophagy, but it is not reflected by COX1 and TIM23 data (Fig. 1E). Interestingly, OPTN was unchanged in Non Tg and G93A in Fig. 5C and D.

Answer: Since in the canonical pathway OPTN and p62 are downstream of Parkin in mitochondrial protein quality control, we had expected a decline of these two mitophagy adaptor proteins on mitochondria in the absence of Parkin. Since the difference between G93A and PKO/G93A was significant but small at 130 days and persistent at end stage (new Expanded View figure 3), we deemed that the other E3-Ub ligases resident in mitochondria, such as March5 and Mull1, are compensating for PKO. Since the mitochondrial levels of these E3-Ub ligases are not increased, we also postulate that is their activity that increases. In addition, as we mention in discussion, it is also likely that other, still uncharacterized, E3-Ub ligases may be upregulated in response to PKO. The field of mitochondrial protein quality control is rapidly expanding and new players are constantly being identified. As we now show in Expanded View figure 3, at end stage there is a massive increase of p62 and OPTN localization in mitochondria. We agree that we do not have all the links yet, and more work on the basic biology of mitochondrial quality control in animal tissues needs to be done. However, we believe that overall the results strongly suggest that there is a component of non-Parkin dependent elimination of mitochondria that takes over when mitochondrial damage reaches a critical stage.

Question 2:

The mitochondrial inner membrane protein markers selected to support the hypothesis that there was increased mitophagy (COX1 and TIM23) also raises the doubt because mitokeima data (Fig.2) shows high level of mitophagy in 60 and 90d old mice, but was reflected only at 120d when the protein levels were considered. Of course, these are two different approaches, but if they don't complement each other it is better to consider only one of them, or use some other mitochondrial marker, for example Cyclophilin D, which is a matrix resident protein. The use of TIM23 is not ideal since the mitochondrial import apparatus has been reported to be affected in ALS. In addition, the use of COX1 raises further doubts because though in Non-TG mice, mitophagy is apparently increasing age dependently (Fig.2C), this was not reflected in the COX1 levels (where it was age

dependently increased, Fig.1E). One explanation is simultaneous increased mitochondrial clearance and generation. But that should be supplemented, at least, with PGC1 α level.

Answer: The reviewer raised an interesting and important point regarding mitochondrial biogenesis. Therefore, as suggested by the reviewer, we have measured spinal cord PGC1 α mRNA at all time points analyzed in the SOD1-G93A and Non Tg mice (added as Fig. 1G and shown here). We detect

an overall increase of PGC1 α mRNA in both genotypes between age 30 and 60 days, suggesting that increased mitochondrial biogenesis is also developmentally regulated. However, at 90 and 120 days PGC1 α mRNA starts declining in G93A SOD1 mice, suggesting an impairment in mitochondrial biogenesis. Thus, increased mitochondrial turnover is compensated in the early stages of disease by biogenesis, but it is not sustained in the symptomatic phases of the disease, which together with the higher degradation rates by mitophagy leads to a depletion, detected by lower levels of mitochondrial proteins, like Tim23 and COXI (Figures 1C, D, E and F).

The other intriguing finding prompted by the reviewer's suggestions, is on the levels of Cyclophilin D. We looked at Cyclophilin D (CypD) in the spinal cord homogenates of the four genotypes at 130 days and at end stage (shown here), and found a remarkable decline of the protein in both G93A and PKO/G93A mice. This decline was completely dissociated from any other mitochondria marker that

we looked at, both at the protein level and the biochemical activities. Importantly, it was not associated with PKO itself, as PKO actually increased cyclophilin D. Therefore, it appears that Cyclophilin D levels are regulated differently than other mitochondrial proteins. It is possible, that the protein is downregulated to increase mitochondrial calcium capacity in the spinal cord, which we and others have previously reported to be significantly decreased early on in the course of the disease. This possibility will be further investigated, also in the context of different mouse strains, but we think that this investigation is beyond the scopes of this manuscript. We would like to explore this finding further, in a more mechanistic manner. If the reviewer agrees, we would prefer not to add these results to the manuscript at this time.

Question 3:

In the discussion section authors mentioned that the Parkin KO mice does not develop severe neurodegeneration due to compensatory mechanisms. As already pointed out at the beginning, they need to address whether the delay in SOD1G93A mediated phenotypes, is not because of those compensatory pathways and how it is directly modulated by Parkin. In that respect, the use of Parkin conditional KO that was generated by the group of Ted Dawson could be of great help.

Answer: We completely agree with the reviewer that crossing the conditional PKO mouse with the SOD1-G93A mouse would be an important experiment. However, we will have to request it, transfer it, and breed it with the G93A mouse. Here four alleles would be involved for induction. Then, we will have to induce excision and follow the time course. It would probably take approximately two years to obtain viable results. Therefore, we think that this could be a future development of this study. For now, although it is a different approach, we have performed in vitro proof of concept experiments with YFP-Parkin expression in NSC34 motor neuron-like cells, which show that it aggravates the toxicity induced by a mitochondrial ROS generator (mtPQ, see also answer to question 2 of reviewer 1). These data in Extended View 5, support the hypothesis that Parkin can be a modulator of mitochondrial induced cell death, under conditions of mitochondrial damage.

Question 4:

It is notable that the G93A mice starts showing muscle weakness and weight loss from 130 days, when mitophagy levels were found to be unchanged (120d, Fig. 2C). Authors need to address this if they want to conclude that increased mitophagy is involved in the disease progression.

Answer: We apologize for lack of clarity on this point. The mice that develop muscle weakness at 130 days are in the C57BL6 background (congenic with PKO mice) with a slower disease progression and later onset (average age of death is 157 days). The mice in which we show mt-Keima fluorescence (cross of mt-Keima mice and SOD1-G93A C57BL6) had similar onset and progression. The 130 days old C57BL6 G93A mice are still in early symptomatic stage, while the 120 days old B6SJL SOD1-G93A mice are close to end stage. We propose that, as disease progresses, reaching the symptomatic stage, mitophagy starts becoming exhausted and plateaus at levels similar to Non Tg mice. This could be due to a decrease of mitophagy components or because the mitochondrial pool has declined. We acknowledge that the fact that we have used different SOD1-G93A mouse lines to maintain congenic background in the PKO/G93A cross, makes it sometimes difficult to visualize the comparisons across time points and disease stages. Here below, we group in a table the two SOD1-G93A mouse models used in this work for the reviewer's convenience.

	Asymptomatic	Presymptomatic	Onset	End Stage
B6SJL (G93A)	30 days	60 days	90 days	120 days
C57BL6 (G93A, mt-Keima G93A)	NA	NA	130 days	Approximately 157 days
Phenotypes	1. Increased mitophagy flux	1. Moderate Parkin protein decline. 2. Increased mitophagy flux.	1. Decline in PGC1 α mRNA and mitophagy flux. 2. Moderate accumulation of mitophagy adaptors. 3. Decreased Parkin protein levels. 4. NMJ denervation and MN loss. 5. Body weight loss starts to be apparent. 6. Moderate decline in mitochondrial proteins.	1. Decreased mitochondrial enzyme activities, mitochondria proteins and PGC1 α mRNA. 2. Decreased Parkin protein levels. 3. Increased accumulation of LC3 II and mitophagy adaptors in mitochondria.

Question 5:

Mitophagy was not measured and compared in the PKO and PKO/SOD1 G93A groups, at least by measuring the standard mitochondrial protein markers.

Answer: We agree with the reviewer that we have not shown data on mitochondrial protein markers in the PKO and PKO/G93A. Ideally, we would have done the mt-Keima cross with these mice, but the multiple alleles involved make this a very hard proposition. As the reviewer suggested, we have measured COXI and Tim23 protein levels at 130 days and disease end stage for the PKO/G93A mice. The results are presented in Expanded View 4 and also shown here.

Palomo GM et al. Expanded View 4

**Question 6:**

The discussion part does not always support the author's findings. "We also detect accumulation of mitochondrially localized MQC proteins, p62 and OPTN, suggesting that ultimately the ability to eliminate damaged mitochondria through mitophagy in SOD1-G93A spinal cord is compromised." - this statement is not supported by the mitokeima data as there was no difference with the non-TG group on 120d. Then they mentioned, "Together, these results indicate that Parkin depletion is due to increased protein turnover, as a result of hyperactive MQC." This is confusing.

Answer: We apologize for the lack of clarity. We now clarify this concept in the revised discussion. The hypothesis goes back to the interpretation that by 120 days mitophagy is no longer more active than in Non Tg because components of the machinery, including Parkin, become depleted. We also discuss the role of other E3-Ub ligases in mitochondria which are not affected by SOD1-G93A and may sustain similar autophagy initiation in G93A and PKO/G93A spinal cord.

Minor concerns:

1. Maybe it is a misunderstanding, but they mentioned in the first paragraph of result section that the SOD1-G93A mice have an average life span of 129 days, but in the survival graph or in the body weight graph no effect is seen before 140 days.

Answer: We apologize that this point was unclear. The mice that develop muscle weakness at 130 days are in the C57BL6 background (congenic with PKO mice) with a slower disease progression and later onset (death at 157 days). The mice in which we show mt-Keima fluorescence are in the mixed C57BL6 background, with similar onset and progression to C57BL6 G93A mice. Therefore the 130 days old C57Bl G93A mice are still in early symptomatic stage, while the 120 days old B6SJL G93A mice are close to end stage. The table added to the response to reviewer 2 point 4 includes this information.

2. In the Fig. 1A, the non-TG OPTN bands do not reflect the quantification.

Answer: We agree with the reviewer that the changes over time of OPTN on mitochondria are moderate and difficult to appreciate, indicating that this specific adaptor is likely not the most important one in the spinal cord of G93A mice. For this reason, we decided to omit OPTN quantification and instead focus on p62, which shows much stronger signal in figure 1. We come back to OPTN in the C57B6 mice (i.e., PKO and PKO/G93A), to show that at end stage they have a strong increase of OPTN in mitochondria (EV fig. 3B, D).

3. Purity of the mitochondrial fraction is a matter of concern, especially in the LC3 blot where a high level of LC3-I was also present in the mitochondrial fraction.

Answer: We note that mitochondrial fractions were not percoll gradient purified, but rather enriched by centrifugation in sucrose buffer. The reason is that, in our experience, gradient purified fractions typically lose the damaged and smaller mitochondria and select for the larger and more intact mitochondrial subpopulations. Therefore, we used the mitochondrial enrichment approach to visualize proteins in all mitochondria from spinal cord. It is possible that some LC3-I associates with mitochondria and co-precipitates with the mitochondrial pellet. Most importantly, here we focus on the comparison among groups of the levels of LC3-II that associates with the mitochondrial fractions.

4. Somehow the bar graphs (+/- SEM) do not always reflect the statistical significance.

Answer: We agree that sometimes the differences between groups are small. All the statistics was performed in the same way for all the comparisons, using GraphPad, as described in methods. After normalizing all the western blot quantification data relative to Non Tg across the many experiments, as recommended by reviewer 1, we see less variability and thus a smaller error bar in some of the groups.

5. The endogenous SOD1 shows high level aggregates in Fig. 4E, is there any difference between PKO and non-TG? The explanation how Parkin knock down enhance the clearance of the aggregates is not clear. If the authors hypothesize that SOD1 aggregates in mitochondrial membrane act as the seed for clearance they need to show that.

Answer: To answer the reviewer's question we performed t-test between Non Tg and PKO and found no statistically significant differences. Furthermore, to see if mitochondria were the source of the difference, we also did filter trap assays of enriched mitochondria fractions. We found that in both G93A and PKO/G93A there is increased aggregated SOD1 and no difference between these two groups. This is now shown here and in Extended View 2. This result suggests that PKO strongly affects extra-mitochondrial SOD1 aggregation and not intra-mitochondrial aggregation. At this point, we do not know if the difference resides in aggregated protein on the mitochondrial surface that washes out when mitochondria are isolated and go into the cytosolic fraction. The result point to an extra-mitochondrial function of Parkin in regulating the levels of SOD1 aggregation. We have added these data in the result and discussion.

6.Fig. 7C, D. Statistical analysis between Non Tg with the other groups are required, if the authors claim that Parkin KO has a role in determining the PGC1 alpha level.

Answer: We performed statistical analyses to compare Non Tg and PKO and found no difference in PGC1 α expression, suggesting that PKO delays PGC1 α decline only in the presence of SOD1-G93A. This was added to the results.

Referee #3 (Remarks for Author):

The manuscript by Palomo et al., uses a well-characterized model of Amyotrophic Lateral Sclerosis (the mutant SOD1-G93A mouse model) to study the mechanisms underlying activation of the mitochondria quality control (MQC), leading to mitophagy and, eventually, neurodegeneration. The study shows that, in the spinal cord of the SOD1-G93A mice two MQC proteins, p62 and optineurin, are activated leading to mitophagy. Looking at the mechanisms of mitochondria dysfunction, the authors show that Parkin and other mitochondria proteins, namely Drp1, Miro1 and Mfn2 (all ubiquitinated by Parkin) and PGC1alpha are also involved. Contrary to the expectations, crossing of the Parkin KO mice awith the SOD1-G93A mice, protects against neurodegeneration and slows down disease progression in the ALS mice.

This is a well written manuscript from one of the best mitochondria groups. The lab has contributed significantly not only to our understanding of mitochondria biology, but also to the identification of mitochondria pathology and dysfunction in ALS. In fact, while many groups have provided evidence that mitochondria play a crucial role in ALS pathogenesis (and in SOD1-ALS in particular), this group has been at the forefront, leading the ALS field with their significant expertise in mitochondria.

As expected from this group, the biochemical analysis presented follows a rigorous experimental approach and it is based on a solid rationale. The novelty of this manuscript is in (a) the identification of Parkin as-perhaps- a significant player in ALS and (b) the use of the mt-Keima/SOD1-G93A to study mitophagy dynamics in vivo. However, the results are largely descriptive and there are several concerns that greatly diminish the enthusiasm for the work in its current format.

Question 1:

As expected from biochemical and immunological analysis from tissues, most of the data presented that rely on western blot or immunohistochemical analysis and quantification of mitochondria proteins, are not as solid as one would like to see to make definite conclusions. Few examples: (1) Figure 1A & B- The effects on OPTN levels over disease progression are small and there is a high variation of effects over time in the SOD1-G93A mice. While at 120 days, OPTN levels seem to increase (albeit a very modest-but significant increase), at 60 and 90 days they seem to decrease (is the difference between non tg and SOD1-G93A mice significant at 90 days?). One would expect either a gradual increase over time (that follows disease progression) or no change followed by a spike at 120 (as for p62-Figure 1 C-D). The authors do not comment on the lack of a gradual increase over time nor on the small (and inconsistent) effects of disease on OPTN expression.

Answer: We completely agree with the reviewer that the changes over time of OPTN on mitochondria are small and difficult to appreciate (see also response to reviewer 2, point 2), unlike p62 at 120 days of age. We think that this may indicate that OPTN is likely not a major mitophagy

adaptor in the initial response in the spinal cord of G93A mice. For this reason, we decided to remove the OPTN panels from Figure 1, while we kept OPTN in Figure 5 and in EV fig. 3, where we compare the four genotypes, but we mention that it likely does not represent a major player in the initial response to mitochondrial damage induced by mutant SOD1, at least at disease onset. We think that the brisk increase of p62 at 120 days is due to the fact that mitophagy is no longer accelerated, likely due to depletion of components, and therefore the flux of degradation of the mitophagy adaptors, such as p62 (and to a lesser extent OPTN) is decreased. The overall result is the increase in the steady state levels detected by western blot. We clarify this interpretation in results/discussion.

Question 2:

FIG 1 E-F There is a discrepancy between the WB shown in Figure 1E and the quantification shown in Fig 1G. Why do COX1 levels increase in WT-SOD1 at 120 days and, overall, increase over time as compared to control? In Figure 1F, the quantification (COX-1 is normalized by actin that does not seem to change so much in the WB) does not show higher levels of COX1 in 120 days old WT over-expressing SOD1 mice.

Answer: We understand the issue of the apparent mismatch between the representative western blots shown in panel E and the calculated averages and SEM in panel F. In fact, this is a recurrent issue, because we looked at many proteins in many biological replicates (mice). We note that for each group and time point we have analyzed 8 mice (4 males and 4 females) and run the blots in sets, in order to be able to always normalize the intensities by the Non Tg controls. We loaded the same calculated amounts of proteins in each lane, but with protein measurement there is always some variability in loading and that is why we used β -actin as a normalizer. We now show another set that more closely represents the averages, but please note that it is only one set out of 8.

Question 3:

FIGURE 5 C, D, E, F. It is puzzling that, contrary to what shown in Figure 1, the graph in Figure 5D, does not show significantly higher levels of OPTN in G93A mice compared to non-tg mice. Perhaps, as said above, with such small and inconsistent effects, are hard to capture putting in to question the relevance of the findings.

Answer: We apologize for the lack of clarity on this point. As mentioned in the response to reviewer 1 (point 7), the C57Bl6 mice in Figure 5 have a later disease onset than the B6SJL mice. Therefore, the end stage is later (please see also the table that we provided for reviewer 2 point 4). When we looked at p62 and OPTN in the mitochondrial fractions of C57BL6 SOD1-G93A mice at end stage, we observed a strong increase compared to controls. We added this graph in Expanded View figure 3 (also shown above in response to reviewer 1).

Question 4:

It is not clear why there is a mismatch in the time course of the mtKeima/SOD1-G93A mice shown in Figure 2 and all the data related to MQC proteins in Figure 1.

Answer: As mentioned above in response to point 1, we interpret the sudden increase of p62 at 120 days as a consequence of the fact that mitophagy is no longer accelerated, possibly due to depletion of components, and therefore the flux of degradation of the mitophagy adaptors, such as p62 is decreased. The overall result is the increase in the steady state levels detected by western blot. The decline in Tim23 and COX1 is likely due to the fact that mitochondrial biogenesis is not increasing, because of a PGC1 α decline and failure to compensate for the loss of mitochondria. We clarify this interpretation also in response to reviewer 2 (point 2) and in the discussion.

Question 5:

In Figure 4D, since the authors are using a dot-blot analysis, they should include the WT-hSOD1 transgenic mice as control.

Answer: We agree that this is a relevant point, but we and others have published previously that human transgenic WT SOD1 does not form aggregates in spinal cord cytosolic and mitochondrial fractions, unlike the G93A mutant SOD1 (Vijayvergiya et al. J Neurosci, 2005). Therefore, we here we focus on the unexpected effects of PKO in G93A mice, because it is a more novel finding.

Question 6:

The data shown in Figure 6 are interesting. As it relates to Figure 6C, at what time do the authors start seeing NMJ defects and denervation? Does denervation precede or is it concomitant with changes in MQC proteins levels and Parkin?

Answer: We thank the reviewer for suggesting to consider this correlation. As it relates to NMJ denervation, the time course in these G93A mice (B6SJL high copy) has been carefully analyzed by Fisher and colleagues (Fischer et al. Exp Neurol, 2004). They have established the time course of the denervation. They showed that denervation starts at 47 days of age in the B6SJL SOD1-G93A mouse. Therefore, the decline in innervated NMJs in this mouse is parallel with the decrease of Parkin. We added this point to the results.

2nd Editorial Decision

19 July 2018

Thank you for the submission of your revised manuscript to EMBO Molecular Medicine. We have now received the enclosed reports from the referees that were asked to re-assess it. As you will see the reviewers are now supportive and I am pleased to inform you that we will be able to accept your manuscript pending final editorial amendments.

Please submit your revised manuscript within two weeks. I look forward to seeing a revised form of your manuscript as soon as possible.

***** Reviewer's comments *****

Referee #1 (Comments on Novelty/Model System for Author):

Scientific rigor is high. The study is well controlled

Referee #1 (Remarks for Author):

The authors responded to all of my concerns by adding new experiments and several clarification points in the text. IN addition ,their response to other reviewers'comments is also more than satisfactory. In my opinion the study is now ready for publication.

Referee #2 (Comments on Novelty/Model System for Author):

Technical approaches are of high quality and the medical impact of this finding is high. Many works highlighted the potential beneficial effect of increased mitophagy specifically in the context of Parkinson's Disease related neurodegeneration. This work highlights for the first time the potential deleterious effect of exacerbated mitophagy in an in vivo model of neurodegeneration. The finding is therefore novel and of great interest. The model system that was used is adequate.

Referee #2 (Remarks for Author):

The experimental data support the conclusions draw by the author and the revised manuscript addresses all the gaps that were raised. I think this is a very interesting work, which highlights in an in vivo model of neurodegeneration the potential dangerous effect of exacerbated mitophagy. I am overall very positive regarding the manuscript and I do not have further comments.

Corresponding Author Name: Giovanni Manfredi

Manuscript Number: EMM-2018-08888-V2